# Broadening the Scope of Graph Regression: Introducing A Novel Dataset with Multiple Representation Settings

## Abstract

Graph regression is a vital task across various domains, however, the majority of publicly available datasets for graph regression are concentrated in the fields of chemistry, drug discovery, and bioinformatics. This narrow focus on dataset availability restricts the development and application of predictive models in other important areas. Here, we introduce a novel graph regression dataset tailored to the domain of software performance prediction, specifically focusing on estimating the execution time of source code. Accurately predicting execution time is crucial for developers, as it provides early insights into the code's complexity. Furthermore, it also facilitates better decision-making in code optimization and refactoring processes. Source code can be represented syntactically as trees and semantically as graphs, capturing the relationships between different code components. In this work, we integrate these two perspectives to create a unified graph representation of source code. We present two versions of the dataset: `RelSC` (Relational Source Code), which incorporates node features, and `Multi-RelSC` (Multi-Relational Source Code), which treats the graphs as multi-relational, allowing nodes to be connected by multiple edges, each representing a distinct semantic relationship. Finally, we apply various Graph Neural Network models to assess their performance in this relatively unexplored task. Our findings demonstrate the potential of these datasets to advance the field of graph regression, particularly in the context of software performance prediction.

## 1 Introduction

Graph Neural Networks (GNNs)(Scarselli et al., 2008; Micheli, 2009) have demonstrated outstanding performance in processing network data across various real-world applications, ranging from biology to recommendation systems. Their ability to effectively model complex relationships between entities, capture structural dependencies, and incorporate node and edge features has made GNNs an essential tool in a variety of domains. High performance in GNNs is attributed not only to advancements in architectural design (Kipf and Welling, 2016; Hamilton et al., 2017; Veličković et al., 2018; Gasteiger et al., 2018; Zhang and Chen, 2018; Wu et al., 2019; Zhang et al., 2021a; Lachi et al., 2024; Zaghen et al., 2024) but also to the availability of publicly accessible benchmark datasets (Armstrong et al., 2013; Hu et al., 2020a; Morris et al., 2020; Dwivedi et al., 2022; Zhiyao et al., 2024; Huang et al., 2024). These benchmarks have played a crucial role in facilitating research progress by providing standardized datasets and tasks, enabling researchers to evaluate, compare, and improve their models consistently.

However, while the availability of public datasets for node and graph classification has driven rapid advancements across fields such as biology (Zhang et al., 2021b; Bongini et al., 2022), mobility (Jiang and Luo, 2022), social networks Li et al. (2023), and recommendation systems Fan et al. (2019), the same is not true for graph regression tasks. Public datasets for graph regression are predominantly concentrated in specific fields, particularly in Chemistry and Drug Discovery Jiang et al. (2021). These datasets have been instrumental in advancing GNN-based models for applications like molecular property prediction Wieder et al. (2020) and drug-target interaction Zhang et al. (2022). Despite their utility, this narrow focus presents a significant limitation: the exploration of graph regression in other domains remains largely underdeveloped due to the lack of diverse, high-quality datasets.

This scarcity of benchmarks beyond Chemistry and Drug Discovery restricts researchers' ability to fully explore the potential of GNNs in graph regression tasks across other fields. Domains such as finance, transportation, environmental modeling, and even social sciences could greatly benefit from graph regression models, but the absence of appropriate datasets makes it challenging to develop, adapt, and evaluate these models effectively. Addressing this gap is essential for expanding the applicability of GNNs to a broader set of problems, enabling the development of more generalizable models, and pushing the boundaries of graph-based machine learning.

In this paper, we introduce novel graph regression datasets for software performance prediction, specifically focusing on execution time estimation. Accurate execution time prediction provides developers with early insights into code complexity, aiding in optimization (Harrelson, 2017) and refactoring decisions (Lindon et al., 2022; Biringa and Kul, 2024). Our datasets broaden the scope of graph regression tasks and serve as valuable benchmarks for exploring GNN applications in software engineering. Source code is traditionally represented using Abstract Syntax Trees (ASTs) (McCarthy, 1960; Neamtiu et al., 2005; Zhang et al., 2019a; Shi et al., 2021; Samoaa et al., 2022a), Control Flow Graphs (CFGs) (Allen, 1970; Campanoni and Crespi Reghizzi, 2009; Koppel et al., 2022; Mitra et al., 2023), and Data Flow Graphs (DFGs) (Dennis and Misunas, 1974; Davis and Keller, 1982; Kavi et al., 1986; Xie et al., 2022), each capturing different aspects of source code. Inspired by Samoaa et al. (2022b), we enhance ASTs by integrating structural and semantic information from CFGs and DFGs, creating a more expressive representation of source code. To support this methodology, we introduce multiple datasets designed for execution time estimation, each provided in two versions. The first, `RelSC`, consists of relational graphs where nodes and edges encode execution-relevant structural properties of Java programs. This extends the dataset introduced in Samoaa et al. (2022a) by incorporating semantic node features, which were previously absent. The second, `Multi-RelSC`, consists of multi-relational graphs where nodes are connected by multiple relationship types, capturing a more comprehensive view of source code interactions. Multi-relational graph regression datasets are scarce in the literature, making this contribution particularly valuable. Our datasets enable more effective research on graph-based execution time prediction in software engineering, fostering advancements in GNN applications within the field.

## 2 Related Work

**Graph regression dataset.** Several open datasets have been released over the past decades, with a predominant focus on Chemistry and Drug Discovery. For molecular property prediction, datasets such as QM9 Wu et al. (2018) and ZINC Gómez-Bombarelli et al. (2018) are used to predict various properties of small molecules. In the realm of solubility and free energy prediction, datasets like ESOL Li et al. (2022) and Freesolv Mobley and Guthrie (2014) aim to forecast the solubility and free energy of molecules. Similarly, Peptides-struct Dwivedi et al. (2022) is employed to predict aggregated 3D properties of peptides at the graph level. PDBbind Liu et al. (2015) is focused on the study of interactions between proteins and ligands. Toxicity and bioactivity prediction tasks utilize datasets such as ogbg-moltox21 Hu et al. (2020a) and ogbg-moltoxcast Hu et al. (2020a) to assess molecular toxicity and bioactivity. Additionally, datasets like ogbg-mollipo Hu et al. (2020a) are dedicated to lipophilicity prediction, while ogbg-molesol Hu et al. (2020a) is used for solubility prediction. Furthermore, the work by Liu et al. Liu et al. (2022) utilizes monomers as polymer graphs to predict properties such as the glass transition temperature. While significant progress has been made in these domains, there is a growing need for comprehensive benchmarks and datasets in other fields to further advance the state of graph regression tasks across diverse applications.

**GNNs in software engineering.** GNNs have become essential in software engineering, effectively modeling the structured nature of source code (Šikić et al., 2022; Nguyen et al., 2022; Allamanis, 2022; Liu et al., 2023). Prior work Allamanis et al. (2018); Guo et al. (2021); Jain et al. (2021) ASTs with semantic edges for code clone detection. CFGs and DFGs have been successfully applied to vulnerability detection (Zhou et al., 2019; Hin et al., 2022) and clone detection Zhang et al. (2019b), outperforming token-based methods (Li et al., 2017; Russell et al., 2018). Recent studies Rafi et al. (2024) show that integrating multi-level graph representations (ASTs, CFGs, DFGs) improves fault localization and automated program repair. These findings

highlight the versatility and effectiveness in capturing structural and semantic code properties, advancing software engineering research.

**Beyond Graph-Based Models.** Machine learning and deep learning have long played a vital role in software engineering. Transformer-based large language models (LLMs) excel in tasks like code generation and defect prediction by leveraging vast pre-training corpora of source code and natural language (Feng et al., 2020; Chen et al., 2021; Lachaux et al., 2021; Roziere et al., 2021). Unlike graph-based methods, these models capture syntactic and semantic patterns without explicit graph structures, making them effective for code completion, bug detection, and refactoring (Nijkamp et al., 2023; Wang et al., 2021). Additionally, traditional machine learning and deep networks effectively model software runtime behavior by leveraging workload parameters—key metrics such as CPU usage and memory consumption that characterize the performance and resource demands of software workloads Laaber et al. (2021); Ha and Zhang (2019). These findings highlight that AI-driven techniques, even beyond graph-centric approaches, remain powerful tools for optimizing performance and enhancing software development.

## 3 Preliminaries

In this section, we introduce the foundational concepts essential for understanding the core contributions of our work. Specifically, we present three key techniques for representing source code as graphs: the Abstract Syntax Tree (AST), the Control Flow Graph (CFG), and the Data Flow Graph (DFG). These representations form the basis for various program analysis methods and are critical for the discussions that follow.

### 3.1 Abstract Syntax Trees

ASTs McCarthy (1960) offer a hierarchical abstraction of source code, focusing on core programming constructs such as variables, operators, and control structures, while ignoring superficial syntactic details like punctuation. Each node in an AST represents a construct from the source code, with edges defining relationships based on the language's syntax rules. The root typically represents the entire program, and the leaves correspond to basic elements like literals or variable names Neamtiu et al. (2005); Samoaa et al. (2023). The process of building an AST involves parsing the source code according to its grammar, creating a structured representation that supports tasks such as code analysis, optimization, and refactoring Zhang et al. (2019a); Shi et al. (2021); Samoaa et al. (2022a). ASTs are widely used in applications such as static analysis, bug detection, and even machine learning-based techniques for code summarization and generation. To gain a deeper understanding of ASTs, in Listing 1 we report a snippet of code and its AST representation is shown in figure 1.

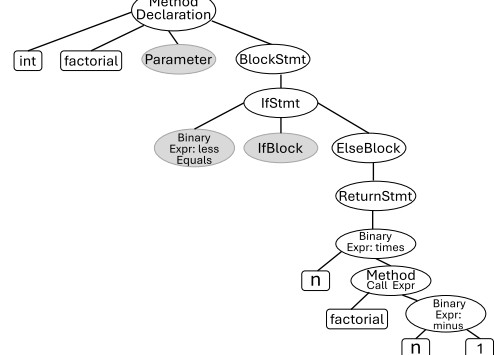

```java
public static int factorial(int n) {
  if (n <= 1) {
    return 1;
  } else {
    return n * factorial(n - 1);
  }
}
```

Listing 1: Simple example of Java source code.

Figure 1: Simplified abstract syntax tree (AST) representing the illustrative example in Listing 1.

## 3.2   Control Flow Graph (CFG)

A Control Flow Graph (CFG) is a directed graph that models the execution flow of a program. Formally, a CFG is defined as a tuple $G_{CFG} = (V, E)$, where $V$ represents a set of basic blocks—sequences of statements with a single entry and exit point—and $E$ denotes directed edges that capture control flow transitions, such as sequential execution, branches, and loops Allamanis et al. (2018). CFGs are widely used in program analysis for tasks such as dead code elimination, path coverage analysis Thomson (2021), vulnerability detection Li et al. (2018), and code summarization Allamanis et al. (2018). A CFG includes a unique entry node marking the program's start and one or more exit nodes representing termination points. Conditional statements introduce multiple outgoing edges, while loops create cycles that model repeated execution. Function calls may extend the CFG into an interprocedural graph, track-

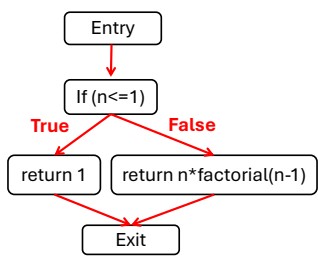

Figure 2: CFG of the method presented in Listing 1

ing function invocations and returns. This structured representation enables precise compiler optimizations, program verification, and machine learning-based code analysis. Figure 2 illustrates the CFG of the 'factorial' method in Listing 1. Execution starts at the "Entry" node and proceeds to the conditional check at "if$(n \leq 1)$" (line 2). If true, execution moves to "return1" (line 3), terminating the function. Otherwise, execution transitions to "$n * \text{factorial}(n-1)$" (line 5), where the recursive call occurs, generating a recursive flow until the base case is reached. All execution paths ultimately converge at the "Exit" node, marking the function's termination. This CFG effectively captures the method's branching logic and recursive structure, illustrating how multiple activations of the function occur before reaching the final return statement.

## 3.3   Data Flow Graph (DFG)

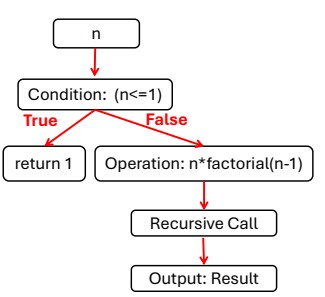

Figure 3: DFG of the method presented in Listing 1

A Data Flow Graph (DFG) is a directed graph that models the flow of data within a program. Formally, a DFG is defined as a tuple $G_{DFG} = (V, E)$, where $V$ represents a set of nodes corresponding to variables or computations, and $E$ denotes directed edges that capture data dependencies, such as variable definitions and their subsequent uses. Unlike CFGs, which represent execution order, DFGs emphasize how values propagate through a program, making them fundamental for static analysis, liveness analysis, and dependency tracking Jiang et al. (2024). They have also been widely applied in machine learning for code property prediction Hellendoorn et al. (2020) and vulnerability detection Li et al. (2018). A DFG consists of nodes representing variable assignments, operations, and function inputs/outputs, with directed edges encoding data flow dependencies. Expressions, arithmetic operations, and function calls contribute to these dependencies, while loops introduce iterative data relationships, and conditionals create multiple propagation paths. By explicitly modeling data flow, DFGs enable precise program optimization, security analysis, and data-driven software engineering. Figure 3 illustrates the DFG of the 'factorial' method in Listing 1. Execution begins at the input node $(n)$, which is evaluated at the comparison node $(n \leq 1)$. If the condition is true, the function returns 1, contributing a constant value node. Otherwise, execution proceeds to compute $n-1$, which is passed to the recursive call factorial$(n-1)$. The result of this call is then multiplied by $n$, forming a data dependency between the recursive output and the final multiplication operation. The computed result is then returned as the function's output. By structuring program execution around data dependencies, DFGs provide a comprehensive view of how values are computed and used, making them essential for compiler optimizations, security verification, and machine learning-based program analysis.

### 3.4 Graph Neural Network

Graph Neural Networks (GNNs) are a type of neural network architecture specifically designed for analyzing graph-structured data. GNNs utilize a mechanism known as message passing, which allows for localized computation across the graph (Gilmer et al., 2017). In essence, the feature vector of each node is iteratively updated by incorporating information from its neighboring nodes. After $l$ iterations, $\mathbf{x}_v^l$ encodes both the structural and attribute information from the $l$-hop neighborhood of node $v$.

More formally, the output of the $l$-th layer of a GNN is defined as:

$$\mathbf{x}_v^l = \mathrm{COMB}^{(l)}(\mathbf{x}_v^{l-1}, \mathrm{AGG}^{(l)}(\{\mathbf{x}_u^{l-1}, u \in N[v]\})) \tag{1}$$

Here, $\mathrm{AGG}^{(l)}$ refers to the aggregation function that gathers features from the neighbours $N[v]$ at the $(l-1)$-th iteration, while $\mathrm{COMB}^{(l)}$ combines the features of the node itself with those of its neighbours. For graph-level tasks such as classification or regression, a global readout function is applied to the node embeddings to produce the final output:

$$\mathbf{o} = \mathrm{READ}(\{\mathbf{x}_v^L, v \in V_G\}). \tag{2}$$

The READ function can be implemented as a sum, mean, or max overall node features or through more sophisticated approaches (Bruna et al., 2013; Yuan and Ji, 2020; Khasahmadi et al., 2020).

Several architectures have been proposedVeličković et al. (2018); Hamilton et al. (2017); Xu et al. (2019); Defferrard et al. (2016), all utilizing the same underlying mechanism but differing in their choice of COMB and AGG functions.

Multi-relational GNNs, such as Relational Graph Convolutional Networks (Schlichtkrull et al., 2017), are specifically designed to handle graphs with multiple types of relations between nodes. In this framework, the message passing mechanism is extended to account for relation types, ensuring that information from different relations is treated distinctively. For a multi-relational graph $G = (V, E, R)$ where $R$ is the set of relation types, the feature update for a node $v \in V$ in the l-th layer is defined as:

$$\mathbf{x}_v^l = \sigma \left( \sum_{r \in R} \sum_{u \in N_r(v)} \frac{1}{c_{r,v}} \mathbf{W}_r \mathbf{x}_u^{l-1} + \mathbf{W}_0 \mathbf{x}_v^{l-1} \right) \tag{3}$$

where $N_r(v)$ represents the neighbors of node $V$ connected by relation $r$, $\mathbf{W}_r$ is a learnable weight matrix specific to relation $r$, $c_{r,v}$ is a normalization constant that can account for the degree of nodes, $\mathbf{W}_0$ is a weight matrix for the self-loop connection, and $\sigma$ is a non-linear activation function. In this formulation, the feature propagation process aggregates messages from neighbors for each relation type separately, applying distinct transformations before combining them. This mechanism allows the model to learn relation-specific patterns, making it particularly suitable for tasks such as knowledge graph completion and multi-relational node classification. Additionally, a global readout function READ can be applied to obtain graph-level outputs as described in Equation 2. Recent advancements in RGCNs have improved multi-relational data modeling Zhu et al. (2019); Yun et al. (2019); Hu et al. (2020b); Lv et al. (2021); Yu et al. (2021); Mitra et al. (2022); Ferrini et al. (2024a;b), yet diverse benchmarks remain limited. This article introduces a dataset and framework to convert Java source code into relational and multi-relational graphs, capturing structural and semantic aspects. Focused on software performance prediction, it offers a novel benchmark for RGCNs in underexplored domains.

## 4 Proposed Datasets

The proposed dataset focuses on predicting the execution time of Java source code, providing an early estimate of code complexity. This is particularly valuable when using cloud computing services, where execution time plays a critical role. The dataset consists of Java code files paired with their corresponding execution times. Each file is parsed into an AST, which is then augmented with edges representing control and data flows, offering a comprehensive view of both code structure and behaviour.

Table 1: Overview of the OSSBuilds and HadoopTests datasets.

| | Project | Description | Files | Avg. Nodes |
|---|---|---|---|---|
| **OSSBuilds** | systemDS | Apache Machine Learning system for data science lifecycle | 127 | 871 |
| | H2 | Java SQL DB | 194 | 2091 |
| | Dubbo | Apache Remote Procedure Call framework | 123 | 616 |
| | RDF4J | Scalable RDF processing | 478 | 450 |
| | **Total** | | **922** | **875** |
| **HadoopTests** | Hadoop | Apache framework for processing large datasets on clusters | **2895** | **1490** |

## 4.1 Data Collection

For our experiments, we employed two different real-world datasets of performance measurements across diverse software environments. The first dataset (*OSSBuild*) consists of actual build data sourced from the continuous integration systems of four open-source projects, representing real-world software development workflows. The second dataset (*HadoopTests*) consists of a larger collection of performance measurements obtained by systematically executing the unit tests of the Hadoop open-source project multiple times in a controlled environment. A summary of both datasets can be found in Table 1. By using datasets from two distinct sources—one capturing variability in real-world build environments (*OSSBuild*) and the other collected in a controlled setting (*HadoopTests*)—we seek to provide an evaluation that considers both real-world complexity and controlled settings. To further address the diversity, and representativeness of our datasets, as well as the steps taken to mitigate potential biases in the data collection process, we provide a detailed analysis in Appendix G. In the following subsections, we provide further details about each dataset used in our experimental studies.

### 4.1.1 OSSBuild Dataset

This dataset, initially utilized in Samoaa et al. (2022b), contains data on test execution times from production build systems for four open-source projects: systemDS [1], H2 [2], Dubbo [3], and RDF4J [4]. These projects utilize public continuous integration servers, from which we extracted test execution times as a proxy for performance during the summer of 2021. Table 1 (top) presents basic statistics about the projects in this dataset. "Files" indicates the number of unit test files for which we collected execution times, and each file will be represented as one graph, while "Avg.Nodes" relates to the average number of nodes in the resulting graphs. Prior to parsing, code comments were removed to reduce the number of nodes in each graph, as they are considered non-essential.

### 4.1.2 HadoopTests Dataset

To overcome the limitations of the OSSBuild dataset, particularly the limited number of files (graphs) per project, we compiled a second dataset for this study. We chose the Apache Hadoop framework [5] due to its extensive number of test files (2,895) and its sufficient complexity. Each unit test in the project was executed five times, with the JUnit framework Samoaa and Leitner (2021) recording the execution duration for each test file at millisecond granularity. The data collection was conducted on a dedicated virtual machine within a private cloud environment equipped with two virtual CPUs and 8 GB of RAM. Following best practices in performance engineering, we disabled all non-essential services during the test runs. Statistics for the HadoopTests dataset are provided in Table 1 (bottom).

---

[1]https://github.com/apache/systemds
[2]https://github.com/h2database/h2database
[3]https://github.com/apache/dubbo
[4]https://github.com/eclipse/rdf4j
[5]https://github.com/apache/hadoop

### 4.2 AST Construction

To construct the AST, we parse the Java code using javalang[6], a pure Python library designed for Java parsing. This parser extracts structural elements of the code while omitting purely syntactical components such as comments, brackets, and code location metadata. The javalang parser produces ASTs by assigning each parsed element to one of 72 predefined node types. These node types represent different program components, such as method declarations, variable assignments, and control flow structures (detailed in Appendix C). Since javalang is widely used in software engineering research, its node type definitions follow a standardized approach, ensuring consistency with existing parsing methodologies. Once the AST is constructed, it forms a tree-like structure (an acyclic undirected graph) composed of these 72 node types. To incorporate this representation into our model, we encode each node type using one-hot encoding, enabling the use of node embeddings for downstream learning tasks.

### 4.3 From AST to `RelSC`

The AST obtained from a Java source code file is initially an acyclic, undirected graph. To transform it into a more expressive representation, we first convert it into a directed graph by assigning directed edges from parent nodes to child nodes. To further enrich the graph and capture both structural and semantic relationships, we introduce 11 additional edge types. These edges integrate information from the AST, CFG, and DFG, enhancing the representation of execution semantics and dependencies within the code. The introduced edges are categorized as follows:

**AST-Derived Edges:** These edges directly preserve the hierarchical structure of the AST.

- **AST Edges** (a): These edges are inherited directly from the AST, maintaining the parent-child relationships within the syntax tree.

- **Next Token** (b): Connects leaf nodes sequentially, capturing the linear order of tokens in the source code.

- **Next Sibling** (c): Links each node to its adjacent sibling in the AST, preserving structural locality.

**Data Flow Edges:** These edges capture dependencies based on variable usage and data propagation.

- **Next Use** (d): Connects a variable node to the next occurrence where it is used, effectively modeling data dependencies between statements.

**Control Flow Edges:** These edges simulate execution paths and conditional branching within the program.

- **If Flow** (e): Connects the predicate (condition) of an if-statement to the corresponding block of code executed when the condition is true.

- **Else Flow** (f): Links the predicate of an if-statement to the alternative (optional) else-block, capturing branching behavior.

- **While Execution Flow** (g): Connects the condition of a while loop to its body, modeling the repeated execution of loop iterations.

- **While Next Flow** (h): Links the last statement inside a while-loop body back to the condition node, simulating the loop execution process.

- **For Execution Flow** (j): Connects the loop condition in a for-statement to the body of the loop, ensuring proper modeling of iterative execution.

---

[6]https://pypi.org/project/javalang/

- **For Next Flow** (k): Similar to the While Next Flow edge, this edge models the execution order within for-loops.

- **Next Statement Flow** (i): Represents the sequential execution of statements within a code block by connecting each statement to the next one in order.

By integrating these edges, the graph effectively captures both syntactic structure and execution behavior, creating a richer representation for downstream tasks such as execution time prediction and performance analysis.

In Figure 4 (left), we present the `RelSC` graph generated from the example in Listing 1. While our approach builds upon the `RelSC` representation introduced in Samoaa et al. (2022b), it incorporates several key enhancements. Most notably, we integrate semantic node type information, which was not considered in Samoaa et al. (2022b). These node types are extracted using the javalang parser, as detailed in Section 4.2, enriching the graph representation with additional syntactic context. Furthermore, unlike Samoaa et al. (2022b), where node embeddings rely solely on structural properties, our approach enhances node feature representation by leveraging both node type encoding and edge information. Given that the `RelSC` graph is a multigraph—where multiple edges can exist between the same pair of nodes—we construct node features by concatenating the one-hot encoding of node types with the summed one-hot encoding of their outgoing edges. This allows for a more expressive representation of both node roles and their relational context within the graph. These improvements make our approach more semantically aware and structurally enriched compared to Samoaa et al. (2022b), ultimately leading to a more informative graph representation for downstream tasks.

### 4.4 From `RelSC` to `Multi-RelSC`

Once `RelSC` graphs have been computed, we also provide a multi-relational version of the dataset, referred to as `Multi-RelSC`. This extension introduces an additional layer of semantic information by categorizing nodes based on their roles and meanings within the Abstract Syntax Tree (AST) (see Section 4.2). The decision to split node types into categories stems from the need to capture the diverse and domain-specific

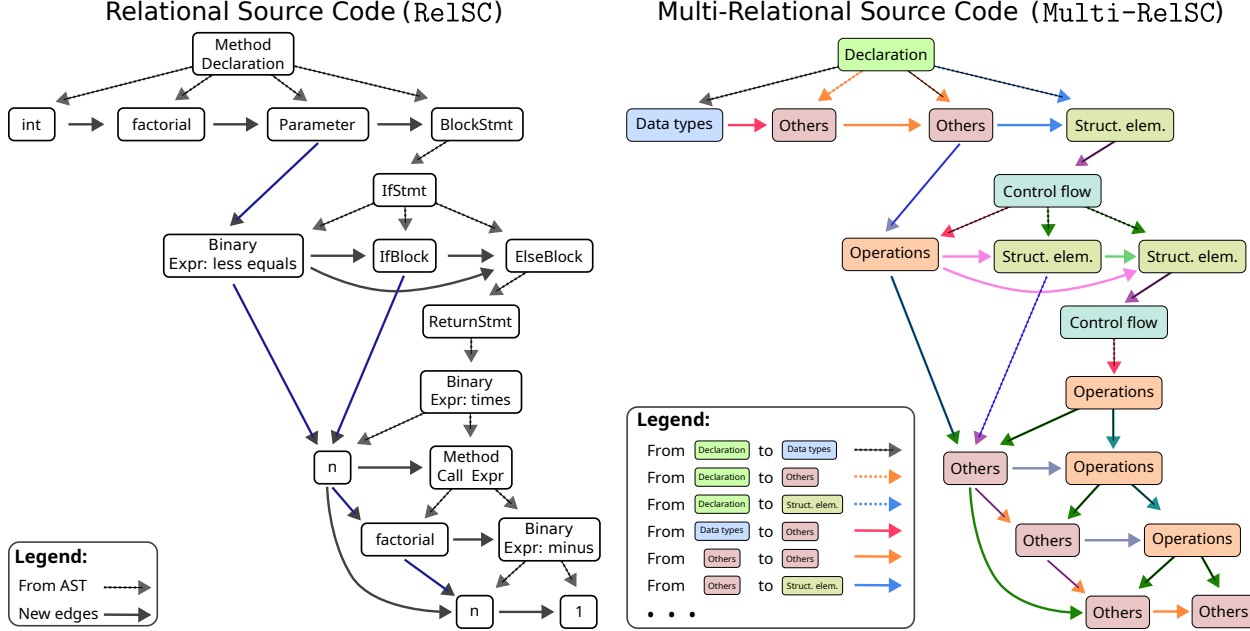

Figure 4: (**Left**) `RelSC` graph for the example presented in Listing 1. (**Right**) `Multi-RelSC` graph for the example presented in Listing 1

relationships that exist in programming constructs. Specifically, we identify seven categories of nodes: **Declarations**, which refer to the definition or declaration of variables, methods, classes, and similar constructs; **Data Types**, representing specific data types or references to types; **Control Flow**, which includes terms associated with constructs that control the program's execution flow; **Operations**, referring to terms that signify operations or expressions; **Structural Elements**, covering structural components of the code such as blocks, compilation units, and packages; **Exceptions and Errors**, relating to exception and error handling mechanisms; and finally, **Others**, for terms that do not fit into any of the previously defined categories. In Appendix C, we provide the categorization of each node type, grouping them into these distinct categories. Additionally, we define a relationship for every possible connection between these categories, resulting in a maximum of 49 possible unique relations (more details in Appendix E). As a result, we construct a multi-relational graph with up to 49 distinct relation types. Each node is represented by a feature vector that combines the one-hot encoding of its node type with the summed one-hot encodings of its outgoing edge types (see Section 4.3). Figure 4 (right) illustrates the `Multi-RelSC` graph corresponding to the example in Listing 1.

## 5 Datasets Statistics

In this section, we provide a detailed analysis of the `RelSC` and `Multi-RelSC` datasets, highlighting their key structural characteristics and diversity. By examining node and edge statistics, as well as node type distributions, we demonstrate the complexity and variability of the datasets. These insights establish the suitability of `RelSC` and `Multi-RelSC` as robust benchmarks for evaluating graph-based models in diverse scenarios and application domains.

`RelSC:` Table 2 summarizes the key characteristics of the homogeneous graphs in our `RelSC` dataset, offering insights into their diversity and complexity. The average node and edge counts vary notably across datasets, with Hadoop having the highest averages, indicating greater complexity, while Dubbo represents a more compact framework, highlighting the dataset's versatility in covering both large-scale and smaller graphs. Variability, as shown by STD values, is significant in H2 and Hadoop, pointing to diverse structural complexities. For instance, Hadoop ranges from 23 to 32,592 nodes and 80 to 127,822 edges, illustrating the presence of both simple and highly complex graphs. RDF4J and SystemDS also show broad ranges, reflecting the dataset's overall diversity. These statistics demonstrate the `RelSC` dataset's suitability as a strong benchmark for evaluating graph-based models, ensuring that GNNs can be tested across different scenarios. The variety of graphs presents challenges and opportunities for developing more sophisticated algorithms that generalize across multiple domains and software systems.

`Multi-RelSC:` Table 2 presents an overview of the `Multi-RelSC` dataset, which consists of multi-relational graphs. Compared to `RelSC`, `Multi-RelSC` contains graphs with a higher average number of edges, such as Hadoop's 11,764.1 edges, indicating a greater degree of connectivity. H2 in OssBuilds has the highest mean node and edge counts, representing the largest graphs in the dataset. The dataset exhibits considerable variation in graph sizes, with Hadoop ranging from 23 nodes and 176 edges to 32,592 nodes and 259,820 edges, demonstrating a broad range of structural characteristics. `Multi-RelSC` offers a collection of graphs, fostering the development of advanced algorithms to address complex software systems.

### 5.1 Distribution of Node Types

Figure 5 shows the node category distributions for `Multi-RelSC` OssBuilds (left) and `Multi-RelSC` Hadoop (right) datasets. Most nodes fall into *"Operation"* and *"Others"*, indicating a high occurrence of expressions, operations, literals, and constants. The standard error (black arrows) is especially large for these categories, particularly in Hadoop, showing high variability across samples. Categories like *"Control Flow"* and *"Data Types"* have lower counts and variability, reflecting the diverse complexity of the graphs. More node distributions are in Appendix D.

| | Hadoop | | OssBuilds | | | | | | | | | | Tot | |
| | | | H2 | | Dubbo | | rdf | | SystemDS | | | | | |
| | $|V|$ | $|E|$ | $|V|$ | $|E|$ | $|V|$ | $|E|$ | $|V|$ | $|E|$ | $|V|$ | $|E|$ | | | $|V|$ | $|E|$ |
|---|---|---|---|---|---|---|---|---|---|---|---|---|---|---|
| mean | 1490.3 | 5731.1 | 2091.3 | 8019.6 | 616.1 | 2354.2 | 449.9 | 1740 | 871.3 | 3321 | | | 875.5 | 3361 |
| std | 2283.4 | 8817.9 | 2631.1 | 10133.8 | 998.9 | 3818.5 | 726.2 | 2826.1 | 629.9 | 2410.9 | | | 1524.7 | 5869.7 |
| min | 23 | 80 | 130 | 500 | 7 | 20 | 22 | 76 | 22 | 78 | | | 7 | 20 |
| max | 32592 | 127822 | 15947 | 61758 | 6374 | 24540 | 5918 | 23146 | 3396 | 13208 | | | 15947 | 61758 |
| mean | 1490.3 | 11764.1 | 2091.3 | 16517.8 | 616.1 | 4811.6 | 449.9 | 3573.6 | 871.3 | 6804.5 | | | 875.5 | 6907.4 |
| std | 2283.4 | 18052.4 | 2631.1 | 20828.4 | 998.9 | 7800.6 | 726.2 | 5783.4 | 629.9 | 4946.3 | | | 1524.7 | 12060.3 |
| min | 23 | 176 | 130 | 1020 | 7 | 40 | 22 | 156 | 22 | 156 | | | 7 | 40 |
| max | 32592 | 259820 | 15947 | 127032 | 6374 | 50672 | 5918 | 47284 | 3396 | 27740 | | | 15947 | 127032 |

Table 2: Statistics for `RelSC` datasets (upper) and for `Multi-RelSC` (lower)

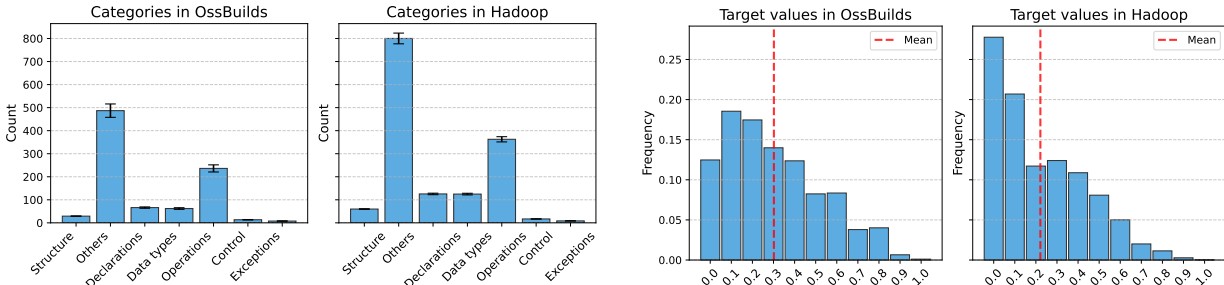

Figure 5: Distribution of node categories in OssBuilds (left) and Hadoop (right).

Figure 6: Distribution of target values in OssBuilds (left) and Hadoop (right).

## 5.2 Target values

Figure 6 illustrates the distribution of target values for OssBuilds (left) and Hadoop (right). The figure shows that both projects contain a higher proportion of fast-executing Java scripts compared to slower ones. The original execution time values range from 0.5 seconds to 4751.51 seconds in OssBuilds and from 0.2 seconds to 1059.67 seconds in Hadoop. Since these values span different ranges across datasets, direct comparisons would be challenging. To ensure comparability, we normalize the target values to the [0,1] range independently for each dataset. Target value distributions for SystemDS, H2, Dubbo, and RDF4J are provided in Appendix H.

# 6  Experiments

In this section, we present the performance of basic GNN and HeteroGNN models on the `RelSC` and `Multi-RelSC` datasets. It is important to note that the main objective of our work is to introduce a novel dataset, not to propose a new architecture.

## 6.1  Implementation Details and Evaluation

We evaluate our models using architectures specifically designed for ASTs, source code, and graphs, ensuring a fair comparison across different architectural paradigms. The AST-based architecture includes Code2Vec Alon et al. (2019), while source code architectures encompass CodeBERT Feng et al. (2020). For graph-based architectures, we consider GCN Kipf and Welling (2017), ChebConv Defferrard et al. (2017), GIN Xu et al. (2019), GraphSAGE Hamilton et al. (2017), and PNA Corso et al. (2020) for `RelSC` graphs. For `Multi-RelSC` datasets, we employ GraphSAGE and GAT Veličković et al. (2018). Notably, for mod-

els trained on `Multi-RelSC` datasets, we leverage heterogeneous message passing[7], which allows the use of distinct parameter sets for different relation types.

All models have two convolutional layers (hidden dimension of 30) and two fully connected layers. We applied mean and max global pooling for graph prediction, with batch normalization and dropout for regularization. Models were implemented using PyTorch-Geometric. Each dataset was split into 70% training, 15% validation, and 15% test sets. It is worth mentioning that, since the primary focus of this work is to introduce novel datasets, we did not perform a hyperparameter search. Each model was trained for 100 epochs with early stopping (patience 15), repeated five times with different seeds, a learning rate of 0.01, and batch size of 32. Experiments were conducted on a machine with four NVIDIA Tesla A100 GPUs (48GB), two Xeon Gold 6338 CPUs, and 256GB DDR4 RAM.

The proposed datasets for the graph regression task exhibit a notable imbalance in target values (see section 5.2). For example, in the Hadoop dataset, approximately 50% of the target values fall within the range of $[0, 0.22]$, indicating a significant concentration of samples in this lower range. This imbalance in the targets makes evaluation more challenging. Therefore, we report the Mean Absolute Error (MAE) as our primary metric. However, since MAE does not account for relative errors, we include additional metrics in Appendix B, specifically Root Mean Squared Error (RMSE), Mean Absolute Percentage Error (MAPE), Spearman's rank correlation coefficient, and the Maximum Relative Error (MRE).

**Other Approaches**   We use two well-known software engineering models that do not rely on graph structures: Code2Vec Alon et al. (2019) and CodeBERT Feng et al. (2020). Code2Vec is a neural network model that represents source code as continuous vectors by extracting structural and semantic relationships from ASTs. It encodes code snippets as sets of path-contexts, which are embedded and weighted using an attention mechanism to identify the most relevant features for predicting code properties like method names. The resulting vectorized representation is then passed through a feedforward neural network to predict source code execution time. In contrast, CodeBERT is a pre-trained transformer-based model designed to learn meaningful representations of source code. We use it to extract vectorized representations of code, which are then fed into a feedforward neural network for execution time prediction. Notably, CodeBERT has a 512-token limit, requiring input code truncation. To address this, we use GPT-3.5 Turbo Ye et al. (2023) to shorten the input code while preserving essential information. Both Code2Vec and CodeBERT are trained for 100 epochs with a batch size of 8.

## 6.2   Results

**RelSC:** Table 3 presents the performance of source code-based, AST-based, and GNN-based models on the `RelSC` datasets, evaluated using MAE along with the standard deviation across five different initialization seeds. Across all datasets, GNN-based models consistently outperform source code-based and AST-based models. Notably, PNA achieves the lowest MAE in every dataset, demonstrating superior performance over all other models.

**`Multi-RelSC`:** Table 3 indicates that HeteroGAT tends to achieve lower MAE values compared to HeteroSAGE across the evaluated datasets. This may be attributed to HeteroGAT's capacity to model multi-relational connections in the `Multi-RelSC` datasets, potentially providing a richer contextual representation for predictions. Variation in MAE across datasets is observed. Hadoop, which has a larger node and edge count, exhibits lower MAE values compared to smaller datasets like SystemDS and H2, where MAE values are generally higher, particularly for HeteroSAGE. Additionally, datasets with higher variability, such as SystemDS and H2, show greater fluctuations in MAE, which could indicate challenges in adapting to diverse graph structures. Overall, HeteroGAT appears to perform more favorably in most cases, though differences in graph size seem to influence MAE outcomes. The multi-relational nature of the `Multi-RelSC` datasets may enable HeteroGAT to take advantage of these relational structures in certain scenarios.

---

[7]https://pytorch-geometric.readthedocs.io/en/latest/generated/torch_geometric.nn.conv.HeteroConv.html

Table 3: Test MAE (lower the better) for `RelSC` and `Multi-RelSC` datasets

|  |  | Hadoop | RDF4J | SystemDS | H2 | Dobbo | OssBuilds |
|---|---|---|---|---|---|---|---|
| Source code | CodeBERT | 0.14($\pm$0.11) | 0.12($\pm$0.10) | 0.17($\pm$0.13) | 0.21($\pm$0.12) | 0.18($\pm$0.12) | 0.15($\pm$0.08) |
| AST | Code2Vec | 0.14($\pm$0.01) | 0.17($\pm$0.01) | 0.19($\pm$0.02) | 0.17($\pm$0.02) | 0.21($\pm$0.02) | 0.15($\pm$0.01) |
| `RelSC` | GCN | 0.12($\pm$0.00) | 0.13($\pm$0.00) | 0.07($\pm$0.02) | 0.18($\pm$0.01) | 0.14($\pm$0.02) | 0.14($\pm$0.01) |
|  | Cheb | 0.11($\pm$0.00) | 0.12($\pm$0.01) | 0.08($\pm$0.04) | 0.18($\pm$0.01) | 0.13($\pm$0.00) | 0.15($\pm$0.01) |
|  | GIN | 0.12($\pm$0.01) | 0.12($\pm$0.00) | 0.08($\pm$0.05) | 0.20($\pm$0.01) | 0.14($\pm$0.01) | 0.14($\pm$0.01) |
|  | GraphSAGE | 0.13($\pm$0.00) | 0.13($\pm$0.01) | 0.07($\pm$0.03) | 0.19($\pm$0.01) | 0.12($\pm$0.01) | 0.14($\pm$0.01) |
|  | PNA | **0.09**($\pm$0.01) | **0.09**($\pm$0.01) | **0.06**($\pm$0.00) | **0.17**($\pm$0.01) | **0.10**($\pm$0.01) | **0.11**($\pm$0.00) |
| `Multi-RelSC` | HeteroSage | 0.27($\pm$0.11) | 0.20($\pm$0.05) | 6.22($\pm$5.45) | 4.35($\pm$3.51) | 4.05($\pm$5.60) | 0.58($\pm$0.31) |
|  | HeteroGAT | 0.14($\pm$0.02) | 0.15($\pm$0.01) | 0.31($\pm$0.11) | 1.09($\pm$0.54) | 0.19($\pm$0.09) | 0.18($\pm$0.02) |

### 6.3 Discussion

The results highlight the challenges posed by the proposed datasets and the varying performance of different models. PNA achieves the best results on `RelSC` datasets, while HeteroGAT outperforms HeteroSAGE on `Multi-RelSC` datasets. However, HeteroGAT struggles on smaller datasets, such as SystemDS and H2, indicating potential limitations in handling less complex graphs. Surprisingly, models trained on `RelSC` datasets outperform those on `Multi-RelSC` datasets, despite the richer information provided by multi-relational structures. This suggests an open challenge in designing models that can fully leverage multi-relational data, which warrants further investigation. Moreover, source code and AST-based models underperform compared to

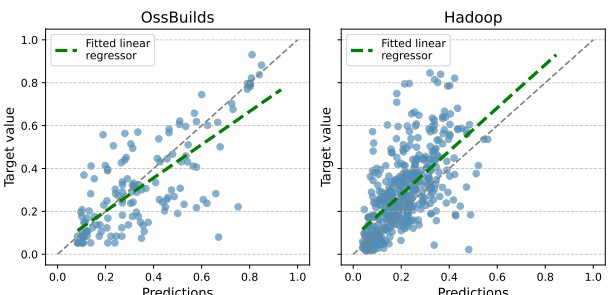

Figure 7: Test predictions versus target values for the PNA model in OssBuilds (left) and Hadoop (right).

GNN models, primarily due to their susceptibility to outliers, as evidenced by the maximum relative error reported in Table 12 (Appendix B). This limitation affects their reliability in execution time prediction tasks, reinforcing the advantages of graph-based representations. These findings establish the proposed datasets as rigorous benchmarks for evaluating GNN models, offering a valuable testbed for developing architectures better suited to real-world graph-based learning tasks. To further illustrate the need for improved models, Figure 7 presents the correlation between predicted and target values for the PNA model on OssBuild (left) and Hadoop (right). The figure reveals significant outliers, particularly in Hadoop, where predictions cluster near zero and fail to estimate values exceeding 0.6. Such inaccuracies can lead to unreliable execution time predictions in real-world applications, emphasizing the necessity for more robust and generalizable models.

### 6.4 Ablation Study

Table 4: Test MAE on OssBuilds using only ASTs

| Model | Test MAE |
|---|---|
| GraphConv | 0.22($\pm$0.02) |
| ChebConv | 0.23($\pm$0.01) |
| GINConv | 0.21($\pm$0.01) |
| GraphSAGE | 0.22($\pm$0.01) |

Abstract Syntax Trees represent source code syntax but lack semantic details like control and data flow. To address this, we augment ASTs with edges from Control Flow Graphs (CFGs) and Data Flow Graphs (DFGs), creating Flow-Augmented ASTs (FA-ASTs). An ablation study on the OssBuilds dataset (Table 4) shows that adding these edges significantly improves performance compared to plain ASTs (Table 3).

The inclusion of flow edges significantly enhances the performance of all models, reducing the test MAE by approximately 0.07 to 0.09. For instance, the MAE for GraphConv improved from 0.22($\pm$0.02) to 0.14($\pm$0.01), ChebConv from 0.23($\pm$0.01) to 0.15($\pm$0.01), GINConv from 0.21($\pm$0.01) to 0.14($\pm$0.01), and GraphSAGE from 0.22($\pm$0.01) to 0.14($\pm$0.01). These results

underscore the critical role of semantic augmentation, as the incorporation of control and data flow information enables GNN models to learn richer representations that better capture execution pathways and dependencies within the code, ultimately leading to significant improvements in prediction accuracy. This demonstrates the importance of flow augmentation for constructing informative graph representations in software performance prediction tasks.

## 7    Real-World Applications

Accurately predicting source code execution time is essential for optimizing software performance, improving development workflows, and enhancing user experience. The proposed datasets, `RelSC` and `Multi-RelSC`, can be leveraged in several impactful ways:

- *Code Optimization and Refactoring:* Modern software development relies heavily on execution time analysis to optimize performance. For instance, Facebook's TAO system dynamically adjusts caching strategies based on execution predictions, improving query response times Bronson et al. (2013). Similarly, Google's Chrome team leverages performance models to prioritize rendering optimizations, enhancing user experience Harrelson (2017).

- *Continuous Integration and Deployment (CI/CD):* Detecting performance regressions early in the development cycle is crucial for maintaining efficient software systems. Large-scale CI/CD platforms, such as those used by Microsoft and Netflix, incorporate performance regression testing to identify slowdowns before deployment. Reliable execution time estimation enables automated detection of inefficient code changes, preventing costly degradations Lindon et al. (2022); Biringa and Kul (2024).

- *Performance-Aware Scheduling:* Effective scheduling in cloud computing relies on accurate estimations of execution time to allocate resources efficiently and minimize delays. Cloud computing platforms such as AWS Lambda and Google Cloud Functions must schedule and allocate resources dynamically Jia et al. (2018); Saravanan et al. (2021); Belgacem (2022).

These applications demonstrate the value of our datasets in driving performance-focused decision-making in software engineering, with potential for future integration into automated performance tuning, debugging, and energy-efficient coding tools.

## 8    Data Release

To facilitate further research, we publicly release the raw data and PyTorch Geometric graph objects on Zenodo, along with the code repository on GitHub[8]. The repository contains model implementations, graph construction instructions, a tutorial for loading the dataset and training models, and dataset statistics. The PyTorch Geometric graph objects include predefined train (70%), validation (15%), and test (15%) splits to ensure consistency across experiments. Since OssBuilds consists of multiple projects (SystemDS, H2, Dubbo, and RDF4J), each individual project follows the same 70%-15%-15% partitioning. Importantly, the train, validation, and test sets of each project are fully contained within the corresponding splits of the complete OssBuilds dataset, ensuring a consistent evaluation framework at both the project-specific and dataset-wide levels. This structured partitioning allows for fine-grained analysis while maintaining comparability across different evaluation scales. Comprehensive instructions for accessing and using these data objects are available in the official GitHub repository, which also includes well-documented code to support reproducibility and facilitate ease of use for researchers and practitioners.

## 9    Conclusion

In this work, we have addressed the critical gap in publicly available benchmarks for graph regression tasks by introducing two novel datasets specifically tailored to software performance prediction. Our proposed

---

[8]https://anonymous.4open.science/r/graph_regression_datasets-407E/

datasets, `RelSC` and `Multi-RelSC`, represent Java source code and their corresponding execution times, providing valuable resources for the exploration of GNN models in a new domain—software engineering. These contributions extend the scope of GNN applications beyond the traditionally explored domains of Chemistry and Drug Discovery, enabling researchers to investigate graph regression in software performance and related fields. With our datasets being publicly accessible, we aim to foster further research, providing a standardized benchmark that can drive the development, evaluation, and comparison of GNN models in software engineering and other underexplored areas.

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

## A    Licensing and Ethical Statement

**Licensing:** To construct our dataset, we rely on source code available on GitHub, distributed under the following licenses:

- Hadoop: Apache License, Version 2.0

- H2: MPL 2.0 (Mozilla Public License, Version 2.0) or EPL 1.0 (Eclipse Public License)

- Dubbo: Apache License, Version 2.0

- rdf: BSD-3-Clause License

- SystemDS: Apache License, Version 2.0

We executed the source code and recorded the execution times, as described in Sections 4.1.1 and 4.1.2. The resulting graphs, along with their execution times, are being released under the CC-BY license.

**Ethical Statement:** This dataset is designed to address challenges in graph representation learning, with a particular emphasis on graph regression tasks. While it is not intended for this purpose, there is a possibility that it could be used to enhance models for harmful applications. However, to the best of our knowledge, our work does not directly pose any threat to individuals or society.

## B    Additional Metrics and validation results

In this section we evaluate standard GNN techniques on the proposed datasets. In particular, tables 6, 7 report the test and validation Root Mean Squared Error (RMSE), tables 8, 9 report the test and validation Mean Absolute Percentage Error (MAPE), tables 10, 11 show the Spearman's Rank Correlation Coefficient ($\rho$) for test and validation data, and finally tables 12, 13 show the Maximum Relative Error (MRE).

The MAPE is defined as

$$MAPE = \frac{1}{n} \sum_{i=1}^{n} \frac{y_i - \bar{y}_i}{y_i} \tag{4}$$

where $n$ is the number of observations, $y_i$ is the actual value, and $\bar{y}_i$ is the predicted value.

While the Spearman's Rank Correlation Coefficient is a non-parametric measure of rank correlation and it is defined as:

$$\rho = 1 - \frac{6 \sum d_i^2}{n(n^2 - 1)} \tag{5}$$

where $n$ is the number of observations, $d_i$ is the difference between the ranks of each pair of observations. Note that $\rho$ ranges from -1 to 1, where $\rho = 1$ indicates perfect positive correlation, $\rho = -1$ indicates perfect negative correlation, and $\rho = 0$ indicates no correlation.

Tables 5–13 present the performance metrics across test and validation splits. Specifically, Table 5 reports the MAE on validation splits. Tables 6 and 7 show the RMSE for test and validation splits, respectively. Similarly, Tables 8 and 9 provide the MAPE, while Tables 10 and 11 present Spearman's Rank Correlation Coefficient. Finally, Tables 12 and 13 report the MRE for test and validation splits.

## C    Node Types

In table 14, we report the definition of each node type with their associated category.

Table 5: Validation MAE (lower the better) for `RelSC` and `Multi-RelSC` datasets

| | | Hadoop | RDF4J | SystemDS | H2 | Dobbo | OssBuilds |
|---|---|---|---|---|---|---|---|
| Source code | CodeBERT | $0.12(\pm0.13)$ | $0.12(\pm0.10)$ | $0.18(\pm0.13)$ | $0.16(\pm0.14)$ | $0.13(\pm0.13)$ | $0.12(\pm0.08)$ |
| AST | Code2Vec | $0.13(\pm0.00)$ | $0.16(\pm0.01)$ | $0.16(\pm0.01)$ | $0.12(\pm0.00)$ | $0.22(\pm0.01)$ | $0.14(\pm0.01)$ |
| RelSC | GCN | $0.11(\pm0.00)$ | $0.13(\pm0.01)$ | $0.06(\pm0.02)$ | $0.13(\pm0.00)$ | $0.09(\pm0.01)$ | $0.14(\pm0.00)$ |
| | Cheb | $0.12(\pm0.00)$ | $0.13(\pm0.01)$ | $0.09(\pm0.03)$ | $0.15(\pm0.00)$ | $0.09(\pm0.01)$ | $0.14(\pm0.00)$ |
| | GIN | $0.11(\pm0.00)$ | $0.12(\pm0.00)$ | $0.07(\pm0.04)$ | $0.14(\pm0.01)$ | $0.08(\pm0.01)$ | $0.14(\pm0.00)$ |
| | GraphSAGE | $0.11(\pm0.00)$ | $0.13(\pm0.01)$ | $0.07(\pm0.03)$ | $0.15(\pm0.00)$ | $0.08(\pm0.00)$ | $0.14(\pm0.00)$ |
| | PNA | $0.09(\pm0.01)$ | $0.09(\pm0.01)$ | $0.06(\pm0.00)$ | $0.12(\pm0.01)$ | $0.07(\pm0.01)$ | $0.10(\pm0.00)$ |
| Multi-RelSC | HeteroSage | $0.17(\pm0.04)$ | $0.16(\pm0.01)$ | $1.13(\pm0.41)$ | $1.29(\pm0.58)$ | $0.38(\pm0.34)$ | $0.47(\pm0.24)$ |
| | HeteroGAT | $0.12(\pm0.00)$ | $0.15(\pm0.00)$ | $0.14(\pm0.01)$ | $0.24(\pm0.07)$ | $0.08(\pm0.03)$ | $0.19(\pm0.05)$ |

Table 6: Test RMSE for `RelSC` and `Multi-RelSC` datasets

| | | Hadoop | RDF4J | SystemDS | H2 | Dubbo | OssBuilds |
|---|---|---|---|---|---|---|---|
| Source code | CodeBERT | $0.17(\pm0.10)$ | $0.15(\pm0.03)$ | $0.21(\pm0.12)$ | $0.20(\pm0.09)$ | $0.19(\pm0.06)$ | $0.18(\pm0.11)$ |
| AST | Code2Vec | $0.17(\pm0.01)$ | $0.21(\pm0.01)$ | $0.22(\pm0.02)$ | $0.22(\pm0.03)$ | $0.26(\pm0.01)$ | $0.18(\pm0.01)$ |
| RelSC | GCN | $0.16(\pm0.00)$ | $0.15(\pm0.01)$ | $0.08(\pm0.02)$ | $0.21(\pm0.00)$ | $0.17(\pm0.01)$ | $0.18(\pm0.01)$ |
| | Cheb | $0.15(\pm0.00)$ | $0.15(\pm0.01)$ | $0.09(\pm0.05)$ | $0.21(\pm0.01)$ | $0.17(\pm0.01)$ | $0.19(\pm0.01)$ |
| | GIN | $0.16(\pm0.01)$ | $0.15(\pm0.00)$ | $0.09(\pm0.05)$ | $0.23(\pm0.01)$ | $0.17(\pm0.01)$ | $0.18(\pm0.01)$ |
| | GraphSAGE | $0.17(\pm0.01)$ | $0.16(\pm0.01)$ | $0.09(\pm0.02)$ | $0.22(\pm0.01)$ | $0.17(\pm0.01)$ | $0.18(\pm0.01)$ |
| | PNA | $0.11(\pm0.02)$ | $0.10(\pm0.01)$ | $0.06(\pm0.02)$ | $0.16(\pm0.00)$ | $0.13(\pm0.01)$ | $0.14(\pm0.02)$ |
| Multi-RelSC | HeteroSage | $0.68(\pm0.54)$ | $0.27(\pm0.11)$ | $8.71(\pm8.88)$ | $6.08(\pm4.53)$ | $7.82(\pm12.22)$ | $1.89(\pm1.99)$ |
| | HeteroGAT | $0.21(\pm0.04)$ | $0.18(\pm0.02)$ | $0.43(\pm0.17)$ | $0.97(\pm0.73)$ | $0.32(\pm0.22)$ | $0.24(\pm0.04)$ |

## D   Node Category of the Datasets

In this section, we report the average number of nodes in each category for the remaining datasets: H2, Dubbo, RDF4J, and SystemDS, as shown in Figures 8 to 11. We previously discussed the node distributions for Hadoop and OssBuilds in Section 5.1.

Across these datasets, there is a noticeable consistency in the dominance of the *"Others"* and *"Operation"* categories, which account for a significant portion of the nodes in each dataset. This trend is indicative of the complex and diverse operations and structural elements within these software systems.

While *"Others"* and *"Operation"* categories consistently lead, the distribution among other categories, such as *"DataTypes"* and *"StructuralElements"*, varies between datasets. For instance, SystemDS and RDF4J show a relatively balanced distribution across these additional categories, whereas H2 and Dubbo exhibit higher variability, as reflected by their broader STD bars. This variability suggests that the graphs within each dataset have distinct structural characteristics, further emphasizing the challenges in graph-based model learning.

Overall, these figures highlight the variability and complexity inherent in each dataset, reinforcing the need for flexible and robust models capable of handling diverse graph structures.

Table 7: Validation RMSE for `RelSC` and `Multi-RelSC` datasets

| | | Hadoop | RDF4J | SystemDS | H2 | Dubbo | OssBuilds |
|---|---|---|---|---|---|---|---|
| Source code | CodeBERT | 0.16(±0.08) | 0.15(±0.05) | 0.22(±0.10) | 0.21(±0.11) | 0.13(±0.07) | 0.11(±0.00) |
| AST | Code2Vec | 0.17(±0.00) | 0.20(±0.06) | 0.20(±0.01) | 0.17(±0.00) | 0.26(±0.01) | 0.17(±0.01) |
| RelSC | GCN | 0.17(±0.00) | 0.17(±0.00) | 0.10(±0.02) | 0.17(±0.01) | 0.11(±0.01) | 0.17(±0.00) |
| | Cheb | 0.17(±0.00) | 0.17(±0.00) | 0.11(±0.03) | 0.19(±0.01) | 0.13(±0.01) | 0.17(±0.01) |
| | GIN | 0.16(±0.00) | 0.17(±0.00) | 0.11(±0.03) | 0.18(±0.01) | 0.11(±0.02) | 0.17(±0.01) |
| | GraphSAGE | 0.17(±0.00) | 0.18(±0.00) | 0.10(±0.02) | 0.19(±0.01) | 0.13(±0.00) | 0.17(±0.00) |
| | PNA | 0.12(±0.01) | 0.13(±0.02) | 0.08(±0.00) | 0.16(±0.00) | 0.10(±0.01) | 0.15(±0.02) |
| Multi-RelSC | HeteroSage | 0.35(±0.21) | 0.19(±0.01) | 0.93(±0.51) | 1.43(±0.99) | 0.55(±0.49) | 0.98(±0.68) |
| | HeteroGAT | 0.18(±0.00) | 0.19(±0.00) | 0.18(±0.02) | 0.32(±0.12) | 0.10(±0.03) | 0.27(±0.12) |

Table 8: Test MAPE for `RelSC` and `Multi-RelSC` datasets. We report "-" to indicate that the value diverged.

| | | Hadoop | RDF4J | SystemDS | H2 | Dubbo | OssBuilds |
|---|---|---|---|---|---|---|---|
| Source code | CodeBERT | 0.59(±0.48) | 0.58(±0.41) | 0.59(±0.44) | 0.88(±0.63) | 1.15(±0.83) | 5.12(±2.30) |
| AST | Code2Vec | 0.68(±0.12) | 0.84(±0.10) | 0.33(±0.06) | 0.58(±0.28) | 0.74(±0.36) | 0.68(±0.13) |
| RelSC | GCN | 0.54(±0.02) | 0.78(±0.08) | 0.09(±0.02) | 0.55(±0.07) | 0.73(±0.21) | 0.68(±0.05) |
| | Cheb | 0.58(±0.08) | 0.68(±0.04) | 0.10(±0.05) | 0.60(±0.07) | 0.64(±0.11) | 0.84(±0.06) |
| | GIN | 0.51(±0.01) | 0.64(±0.04) | 0.11(±0.06) | 0.60(±0.07) | 0.70(±0.10) | 0.80(±0.08) |
| | GraphSAGE | 0.59(±0.02) | 0.81(±0.08) | 0.10(±0.03) | 0.65(±0.03) | 0.55(±0.03) | 0.67(±0.05) |
| | PNA | 0.44(±0.05) | 0.51(±0.10) | 0.06(±0.02) | 0.41(±0.08) | 0.42(±0.05) | 0.56(±0.02) |
| Multi-RelSC | HeteroSage | 1.11(±0.25) | 1.18(±0.24) | 7.71(±7.24) | 10.59(±8.28) | 12.59(±18.93) | 2.03(±0.90) |
| | HeteroGAT | 0.67(±0.09) | 0.94(±0.03) | 0.41(±0.14) | 1.73(±1.12) | 0.73(±0.23) | 0.93(±0.06) |

# E   Relations on the Datasets

In this section, we discuss the average number of relations between different node categories for each `Multi-RelSC` dataset. Figures 12-17 show a heatmap where the rows and columns correspond to various categories of nodes (defined in Section 4.4), such as "Declarations," "Control Flow," "Data Types", "Operations", and "Others".

A common pattern across all datasets is the significant number of relations involving the *"Operation"* and *"Others"* categories. These categories consistently show higher interaction counts, indicating their central role in the overall structure of the software systems. Notably, the *"Others"* category frequently interacts with *"Operation"* nodes, underscoring the complexity and interdependence of various node types within the graphs.

The *"Declarations"* and *"Data Types"* categories also show considerable relations, particularly in datasets like H2 and SystemDS (Figures 13 and 17), where they interact heavily with *"Operation"* nodes. This suggests that these systems have a more intricate structure with a higher degree of dependencies between different code elements.

Differences across datasets are most evident in the intensity of specific relations. For example, H2 and Hadoop (Figures 13 and 14) exhibit a higher number of relations between *"Operation"* and *"Others"* compared to Dubbo and RDF4J (Figures 12 and 16), indicating that the former systems have more complex and interconnected codebases.

Overall, these heatmaps illustrate the relational complexity within each dataset, highlighting the critical role of *"Operation"* and *"Others"* categories in maintaining the structural integrity of the codebase. This complexity presents challenges for graph-based models, which must effectively capture these dense interdependencies to make accurate predictions.

Table 9: Validation MAPE for `RelSC` and `Multi-RelSC` datasets. We report "-" to indicate that the value diverged.

| | | Hadoop | RDF4J | SystemDS | H2 | Dubbo | OssBuilds |
|---|---|---|---|---|---|---|---|
| Source code | CodeBERT | $0.97(\pm0.34)$ | $0.74(\pm0.61)$ | $0.58(\pm0.33)$ | $1.61(\pm1.08)$ | $0.26(\pm0.21)$ | $0.73(\pm0.47)$ |
| AST | Code2Vec | $2.20(\pm0.51)$ | $1.27(\pm0.25)$ | $0.28(\pm0.62)$ | - | $0.62(\pm0.12)$ | $0.62(\pm0.12)$ |
| RelSC | GCN | $1.26(\pm0.12)$ | $0.63(\pm0.07)$ | $0.10(\pm0.03)$ | - | $0.54(\pm0.09)$ | $0.59(\pm0.03)$ |
| | Cheb | $1.44(\pm0.17)$ | $0.61(\pm0.04)$ | $0.13(\pm0.04)$ | - | $0.55(\pm0.09)$ | $0.58(\pm0.02)$ |
| | GIN | $1.19(\pm0.08)$ | $0.55(\pm0.02)$ | $0.11(\pm0.04)$ | - | $0.51(\pm0.07)$ | $0.61(\pm0.11)$ |
| | GraphSAGE | $1.32(\pm0.06)$ | $0.67(\pm0.06)$ | $0.11(\pm0.03)$ | - | $0.45(\pm0.01)$ | $0.51(\pm0.10)$ |
| | PNA | $1.01(\pm0.12)$ | $0.51(\pm0.08)$ | $0.08(\pm0.01)$ | - | $0.42(\pm0.05)$ | $0.44(\pm0.09)$ |
| Multi-RelSC | HeteroSage | $1.85(\pm0.53)$ | $0.84(\pm0.05)$ | $1.09(\pm0.60)$ | - | $1.89(\pm1.72)$ | - |
| | HeteroGAT | $1.40(\pm0.15)$ | $0.79(\pm0.05)$ | $0.21(\pm0.01)$ | $1.01(\pm0.09)$ | $0.55(\pm0.11)$ | $0.61(\pm0.08)$ |

Table 10: Test Spearman's Rank Correlation Coefficient ($\rho$) for `RelSC` and `Multi-RelSC` datasets (higher is better).

| | | Hadoop | RDF4J | SystemDS | H2 | Dubbo | OssBuilds |
|---|---|---|---|---|---|---|---|
| Source code | CodeBERT | $0.55(\pm0.23)$ | $0.58(\pm0.21)$ | $0.31(\pm0.09)$ | $0.19(\pm0.11)$ | $0.08(\pm0.02)$ | $0.14(\pm0.03)$ |
| AST | Code2Vec | $0.33(\pm0.07)$ | $0.26(\pm0.06)$ | $0.25(\pm0.26)$ | $-0.12(\pm0.15)$ | $0.03(\pm0.21)$ | $0.48(\pm0.08)$ |
| RelSC | GCN | $0.61(\pm0.03)$ | $0.52(\pm0.03)$ | $0.67(\pm0.04)$ | $0.28(\pm0.09)$ | $0.32(\pm0.32)$ | $0.59(\pm0.03)$ |
| | Cheb | $0.64(\pm0.04)$ | $0.50(\pm0.05)$ | $0.74(\pm0.17)$ | nan | $0.49(\pm0.04)$ | $0.52(\pm0.03)$ |
| | GIN | $0.64(\pm0.03)$ | $0.53(\pm0.02)$ | $0.67(\pm0.08)$ | $0.23(\pm0.09)$ | $0.23(\pm0.35)$ | $0.55(\pm0.05)$ |
| | GraphSAGE | $0.57(\pm0.02)$ | $0.38(\pm0.05)$ | $0.77(\pm0.06)$ | nan | $0.41(\pm0.08)$ | $0.56(\pm0.04)$ |
| | PNA | $0.71(\pm0.02)$ | $0.57(\pm0.01)$ | $0.68(\pm0.00)$ | $0.48(\pm0.05)$ | $0.51(\pm0.00)$ | $0.68(\pm0.03)$ |
| Multi-RelSC | HeteroSage | $0.21(\pm0.21)$ | $0.20(\pm0.07)$ | $-0.34(\pm0.08)$ | $0.02(\pm0.31)$ | $0.13(\pm0.47)$ | $0.24(\pm0.18)$ |
| | HeteroGAT | $0.50(\pm0.11)$ | $0.32(\pm0.07)$ | $0.24(\pm0.31)$ | $0.22(\pm0.27)$ | $0.41(\pm0.17)$ | $0.40(\pm0.04)$ |

Table 11: Validation Spearman's Rank Correlation Coefficient ($\rho$) for `RelSC` and `Multi-RelSC` datasets (higher is better).

| | | Hadoop | RDF4J | SystemDS | H2 | Dubbo | OssBuilds |
|---|---|---|---|---|---|---|---|
| Source code | CodeBERT | $0.62(\pm0.15)$ | $0.56(\pm0.13)$ | $0.44(\pm0.09)$ | $0.67(\pm0.18)$ | $0.33(\pm0.03)$ | $0.81(\pm0.21)$ |
| AST | Code2Vec | $0.43(\pm0.02)$ | $0.40(\pm0.06)$ | $0.43(\pm0.03)$ | $0.28(\pm0.06)$ | $0.36(\pm0.05)$ | $0.52(\pm0.03)$ |
| RelSC | GCN | $0.59(\pm0.03)$ | $0.54(\pm0.02)$ | $0.60(\pm0.14)$ | $0.52(\pm0.06)$ | $0.29(\pm0.03)$ | $0.50(\pm0.03)$ |
| | Cheb | $0.58(\pm0.03)$ | $0.52(\pm0.06)$ | $0.51(\pm0.17)$ | nan | $0.28(\pm0.03)$ | $0.48(\pm0.01)$ |
| | GIN | $0.61(\pm0.02)$ | $0.54(\pm0.02)$ | $0.55(\pm0.18)$ | $0.30(\pm0.34)$ | $0.18(\pm0.05)$ | $0.49(\pm0.02)$ |
| | GraphSAGE | $0.50(\pm0.02)$ | $0.46(\pm0.04)$ | $0.66(\pm0.06)$ | nan | $0.26(\pm0.04)$ | $0.48(\pm0.03)$ |
| | PNA | $0.73(\pm0.01)$ | $0.55(\pm0.02)$ | $0.69(\pm0.01)$ | $0.58(\pm0.01)$ | $0.48(\pm0.04)$ | $0.66(\pm0.01)$ |
| Multi-RelSC | HeteroSage | $0.31(\pm0.09)$ | $0.26(\pm0.05)$ | $-0.10(\pm0.38)$ | $0.21(\pm0.12)$ | $0.07(\pm0.09)$ | $0.19(\pm0.07)$ |
| | HeteroGAT | $0.50(\pm0.05)$ | $0.33(\pm0.05)$ | $0.14(\pm0.23)$ | $0.30(\pm0.08)$ | $0.26(\pm0.11)$ | $0.42(\pm0.05)$ |

Table 12: Test MRE for `RelSC` and `Multi-RelSC` datasets (lower is better)

| | | Hadoop | RDF4J | SystemDS | H2 | Dubbo | OssBuilds |
|---|---|---|---|---|---|---|---|
| Source code | CodeBERT | $42(\pm25)$ | $58(\pm48)$ | $75(\pm61)$ | $83(\pm11)$ | $51(\pm9)$ | $929(\pm81)$ |
| AST | Code2Vec | $3(\pm1)$ | $1948(\pm239)$ | $15(\pm9)$ | $7(\pm4)$ | $5373(\pm629)$ | $1823(\pm148)$ |
| RelSC | GCN | $19(\pm2)$ | $3(\pm0)$ | $2(\pm0)$ | $3(\pm1)$ | $2(\pm1)$ | $5(\pm0)$ |
| | Cheb | $22(\pm3)$ | $3(\pm2)$ | $1(\pm0)$ | $4(\pm1)$ | $2(\pm0)$ | $5(\pm1)$ |
| | GIN | $18(\pm2)$ | $3(\pm0)$ | $1(\pm0)$ | $3(\pm1)$ | $2(\pm1)$ | $6(\pm0)$ |
| | GraphSAGE | $15(\pm3)$ | $4(\pm1)$ | $1(\pm0)$ | $4(\pm0)$ | $1(\pm0)$ | $4(\pm0)$ |
| | PNA | $11(\pm2)$ | $3(\pm0)$ | $1(\pm0)$ | $7(\pm4)$ | $1(\pm0)$ | $3(\pm0)$ |
| Multi-RelSC | HeteroSage | $23(\pm8)$ | $5(\pm1)$ | $33(\pm3)$ | $42(\pm26)$ | $73(\pm11)$ | $30(\pm27)$ |
| | HeteroGAT | $13(\pm2)$ | $3(\pm0)$ | $1(\pm1)$ | $8(\pm4)$ | $3(\pm2)$ | $5(\pm0)$ |

Table 13: Validation MRE for `RelSC` and `Multi-RelSC` datasets (lower is better)

| | | Hadoop | RDF4J | SystemDS | H2 | Dubbo | OssBuilds |
|---|---|---|---|---|---|---|---|
| Source code | CodeBERT | 79(±34) | 61(±21) | 31(±18) | 122(±83) | 13(±11) | 38(±15) |
| AST | Code2Vec | 6819(±814) | 511(±101) | 5953(±363) | 705(±23) | 5366(±226) | 3076(±211) |
| RelSC | GCN | 107(±32) | 5(±0) | 1(±0) | 3(±1) | 2(±1) | 9(±1) |
| | Cheb | 118(±12) | 5(±0) | 1(±0) | 4(±0) | 2(±0) | 9(±2) |
| | GIN | 79(±23) | 6(±0) | 1(±0) | 3(±1) | 2(±0) | 10(±2) |
| | GraphSAGE | 87(±13) | 5(±0) | 1(±0) | 4(±0) | 2(±0) | 7(±1) |
| | PNA | 51(±2) | 3(±1) | 1(±0) | 3(±4) | 1(±0) | 6(±0) |
| Multi-RelSC | HeteroSage | 74(±26) | 4(±1) | 3(±2) | 13(±10) | 5(±3) | 29(±12) |
| | HeteroGAT | 67(±14) | 3(±0) | 1(±0) | 2(±0) | 2(±0) | 14(±3) |

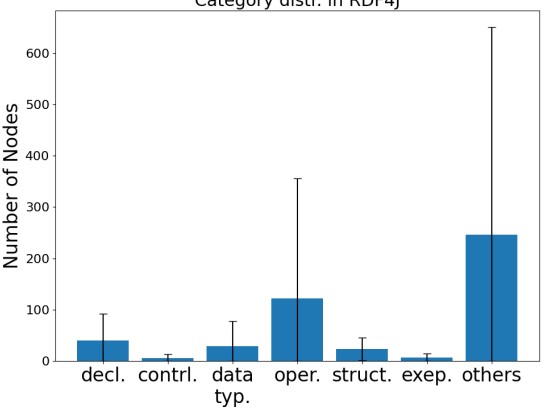

Figure 8: Node Category Distribution for `Multi-RelSC` RDF4J dataset

Figure 9: Node Category Distribution for `Multi-RelSC` SystemDS dataset

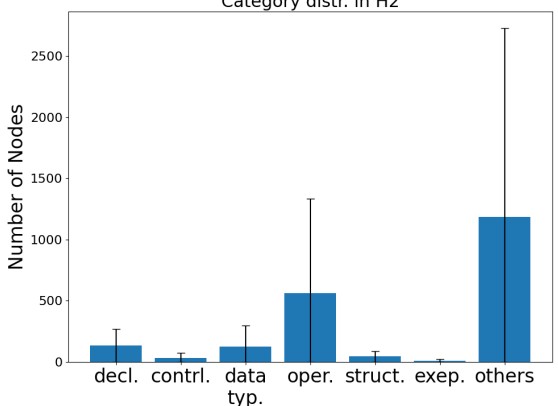

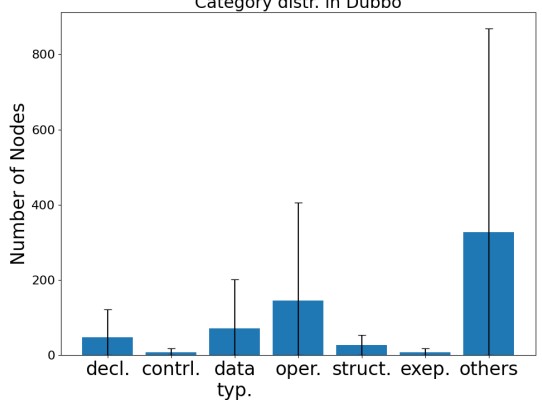

Figure 10: Node Category Distribution for `Multi-RelSC` H2 dataset

Figure 11: Node Category Distribution for `Multi-RelSC` Dubbo dataset

| Node type | Description | Category |
|---|---|---|
| AnnotationMethod | Defines a method used in annotations, often to specify default values for elements | declarations |
| InferredFormalParameter | A formal parameter whose type is inferred by the compiler, often in lambda expressions | declarations |
| LocalVariableDeclaration | Declares a variable within a method, constructor, or block, with local scope | declarations |
| SuperConstructorInvocation | Calls the constructor of the superclass from a subclass constructor | expressions_and_operations |
| Import | Imports classes or entire packages to make them available for use in a Java file | code_structure |
| ArraySelector | Used to select an element from an array using its index | types_and_references |
| BreakStatement | Terminates the nearest enclosing loop or switch statement | control_flow |
| FieldDeclaration | Declares a variable at the class level, which can be accessed by methods of the class | declarations |
| EnumDeclaration | Declares an enumeration, a special Java type used to define collections of constants | declarations |
| ConstructorDeclaration | Declares a constructor, a special method to create and initialize objects of a class | declarations |
| Annotation | A form of metadata that provides data about a program | code_structure |
| ReferenceType | Specifies a type that refers to objects, such as classes, arrays, or interfaces | types_and_references |
| EnhancedForControl | Control structure used to iterate over collections or arrays in a simplified way | control_flow |
| TypeParameter | Represents a generic parameter in a class, interface, or method | declarations |
| Statement | A single unit of execution within a Java program, such as a declaration or expression | control_flow |
| CompilationUnit | Represents an entire Java source file, including package, imports, and class | code_structure |
| EnumConstantDeclaration | Declares constants within an enum type | literals_and_constants |
| IfStatement | A conditional statement that executes code based on a true or false condition | control_flow |
| ClassCreator | Creates an instance of a class, possibly an inner or anonymous class | code_structure |
| SwitchStatement | Selects one of many code blocks to execute based on the value of an expression | control_flow |
| EnumBody | Defines the body of an enum, including constants and other fields or methods | code_structure |
| PackageDeclaration | Declares the package that a Java class or interface belongs to | code_structure |
| Cast | Converts an object or value from one type to another | types_and_references |
| VariableDeclaration | Declares a variable, specifying its type and optional initial value | declarations |
| ArrayCreator | Creates a new array with a specified size and type | types_and_references |
| This | Refers to the current instance of a class | types_and_references |
| MethodReference | Refers to a method by name without executing it, often used in lambda expressions | expressions_and_operations |
| InnerClassCreator | Creates an instance of an inner class | code_structure |
| InterfaceDeclaration | Declares an interface, which can contain method signatures and constants | declarations |
| FormalParameter | Declares a parameter in a method or constructor | declarations |
| CatchClauseParameter | A parameter used in the catch block to represent an exception | exceptions |
| SynchronizedStatement | Ensures that a block of code is executed by only one thread at a time | control_flow |
| VoidClassReference | Refers to the special 'void' type, representing the absence of a return value | types_and_references |
| TypeArgument | An actual type passed as a parameter to a generic type | types_and_references |
| DoStatement | Executes a block of code at least once, then repeatedly based on a condition | control_flow |
| Assignment | Assigns a value to a variable | expressions_and_operations |
| ContinueStatement | Skips the current iteration of a loop and proceeds to the next iteration | control_flow |
| AssertStatement | Tests an assertion about the program, throwing an error if the assertion fails | exceptions |
| ExplicitConstructorInvocation | Explicitly calls another constructor in the same class or a superclass | declarations |
| AnnotationDeclaration | Declares an annotation type, used to create custom annotations | declarations |
| StringLiteralExpr | Represents a literal string value in the code | literals_and_constants |
| PrimitiveType | Represents a primitive data type such as int, char, or boolean | types_and_references |
| TryStatement | Defines a block of code that attempts execution and handles exceptions | control_flow |
| ElementArrayValue | Represents an array of values in an annotation element | code_structure |
| BlockStatement | Groups multiple statements together in a block enclosed by braces | code_structure |
| ClassReference | Refers to a class, often using its fully qualified name | types_and_references |
| ReturnStatement | Terminates a method and optionally returns a value | control_flow |
| IntegerLiteralExpr | Represents a literal integer value in the code | literals_and_constants |
| TernaryExpression | A shorthand conditional expression | expressions_and_operations |
| VariableDeclarator | Declares a variable and its initial value in one statement | declarations |
| BinaryOperation | Represents an operation involving two operands, such as addition or comparison | expressions_and_operations |
| ClassDeclaration | Declares a class, including its name, superclass, and body | declarations |
| TryResource | Represents a resource in a try-with-resources statement that is automatically closed | exceptions |
| MemberReference | Refers to a member of a class, such as a field or method | expressions_and_operations |
| SuperMemberReference | Refers to a member in the superclass of the current class | expressions_and_operations |
| Literal | Represents a literal value, such as a number, character, or boolean | literals_and_constants |
| CatchClause | Handles exceptions thrown in a try block | exceptions |
| WhileStatement | Executes a block of code repeatedly based on a condition | control_flow |
| ElementValuePair | Represents a key-value pair in an annotation | code_structure |
| ForStatement | Defines a traditional for loop with initialization, condition, and iteration | control_flow |
| StatementExpression | Represents an expression that can stand as a statement | expressions_and_operations |
| ConstantDeclaration | Declares a constant, which is a variable whose value cannot be changed | declarations |
| ArrayInitializer | Specifies the initial values for an array | types_and_references |
| MethodInvocation | Invokes a method on an object or class | expressions_and_operations |
| Modifier | Defines modifiers for classes, methods, or fields, such as public, private, or static | declarations |
| ThrowStatement | Throws an exception, signaling an error or abnormal condition | control_flow |
| LambdaExpression | Represents an anonymous function | expressions_and_operations |
| SwitchStatementCase | Represents a case label in a switch statement, matching specific values | code_structure |
| MethodDeclaration | Declares a method, including its return type, name, and parameters | declarations |
| BasicType | Represents a basic data type such as int, float, or char | types_and_references |
| SuperMethodInvocation | Invokes a method from the superclass of the current class | expressions_and_operations |
| ForControl | Specifies the initialization, condition, and update parts of a for loop | control_flow |
| CompilationUnit | Represents the top-level node in AST produced by the parser as the root of the tree | declarations |

Table 14: Conversion table from NodeType to Category

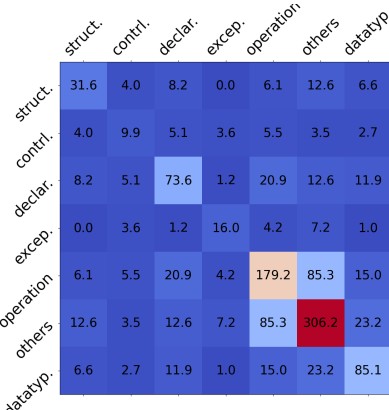

Figure 12: Average number of relations for dataset `Multi-RelSC` Dubbo

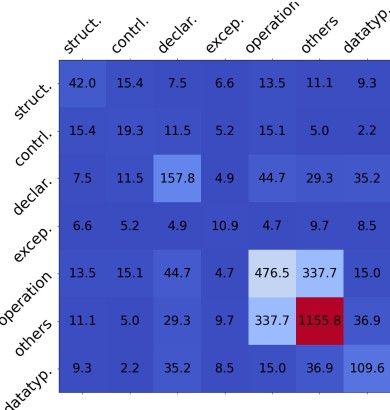

Figure 13: Average number of relations for dataset `Multi-RelSC` H2

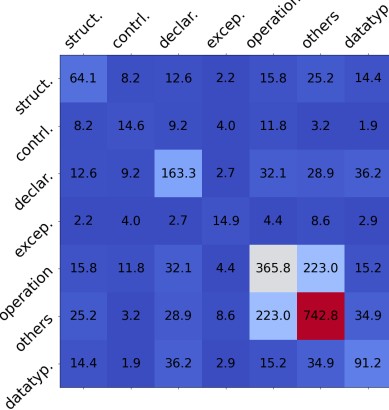

Figure 14: Average number of relations for dataset `Multi-RelSC` Hadoop

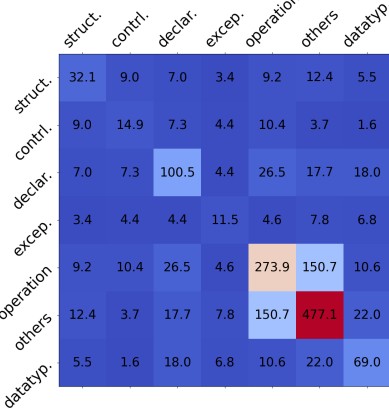

Figure 15: Average number of relations for dataset `Multi-RelSC` OssBuilds

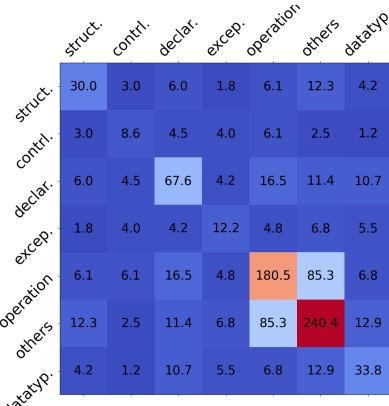

Figure 16: Average number of relations for dataset `Multi-RelSC` RDF4J

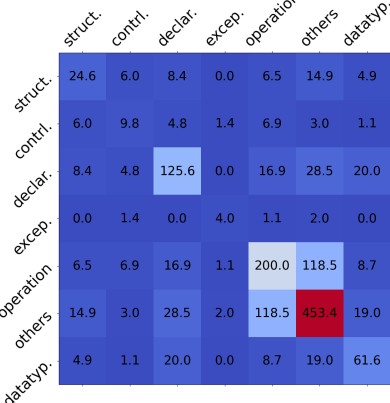

Figure 17: Average number of relations for dataset `Multi-RelSC` SystemDS

## F  Additional Graph Statistics

This section provides additional statistics for an overview of the proposed datasets. Figures 18 and 19 show two `RelSC` and two `Multi-RelSC` networks for Hadoop and OssBuilds, respectively.

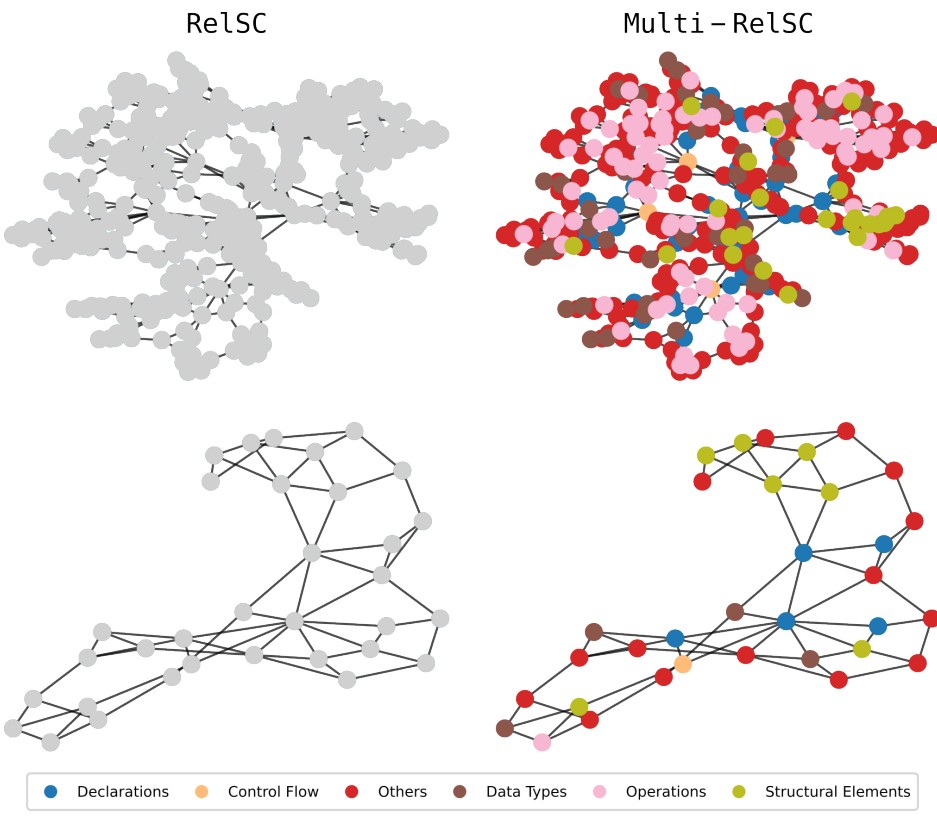

Figure 18: Example of `RelSC` and `Multi-RelSC` graphs from Hadoop

In Table 15, we present the means and standard deviations of several key graph metrics calculated for the proposed datasets. Specifically, we report the average density, indicating the proportion of actual connections to possible connections within each graph. We also include the average degree, reflecting the mean number of connections per node, and the average clustering coefficient, which describes the tendency of nodes to form tightly connected groups. Additionally, we provide the average diameter, representing the longest shortest path between any two nodes, and the average path length, capturing the mean shortest path across all node pairs. Lastly, we report the degree assortativity, which measures the correlation in degree between connected nodes.

| Dataset | Density | Degree | Clustering | Diameter | Path Length | Assortativity |
|---|---|---|---|---|---|---|
| SystemDS | $0.010$ ($\pm$ 0.023) | $3.80$ ($\pm$ 0.06) | $0.29$ ($\pm$ 0.02) | $18.3$ ($\pm$ 4.5) | $7.6$ ($\pm$ 1.3) | $0.12$ ($\pm$ 0.06) |
| Dubbo | $0.026$ ($\pm$ 0.047) | $3.80$ ($\pm$ 0.12) | $0.31$ ($\pm$ 0.04) | $13.9$ ($\pm$ 3.7) | $6.7$ ($\pm$ 1.4) | $0.15$ ($\pm$ 0.09) |
| RDF | $0.041$ ($\pm$ 0.046) | $3.78$ ($\pm$ 0.14) | $0.30$ ($\pm$ 0.03) | $12.7$ ($\pm$ 5.7) | $5.9$ ($\pm$ 2.1) | $0.17$ ($\pm$ 0.08) |
| H2 | $0.005$ ($\pm$ 0.005) | $3.82$ ($\pm$ 0.05) | $0.33$ ($\pm$ 0.02) | $22.1$ ($\pm$ 9.1) | $8.6$ ($\pm$ 1.9) | $0.11$ ($\pm$ 0.08) |
| OSSBuilds | $0.027$ ($\pm$ 0.041) | $3.79$ ($\pm$ 0.12) | $0.31$ ($\pm$ 0.03) | $15.6$ ($\pm$ 7.3) | $6.8$ ($\pm$ 2.2) | $0.15$ ($\pm$ 0.08) |
| Hadoop | $0.011$ ($\pm$ 0.018) | $3.82$ ($\pm$ 0.06) | $0.30$ ($\pm$ 0.02) | $17.3$ ($\pm$ 11.7) | $7.5$ ($\pm$ 3.1) | $0.12$ ($\pm$ 0.07) |

Table 15: Dataset Statistics: Mean Values with Standard Deviations

RelSC                          Multi−RelSC

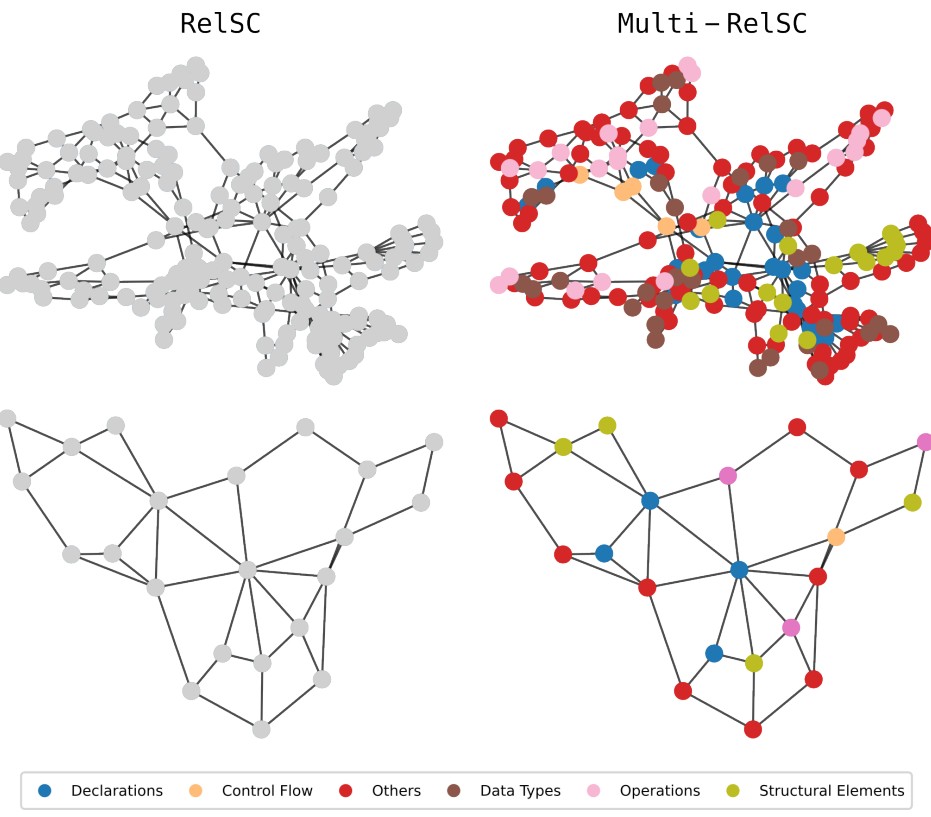

Figure 19: Example of `RelSC` and `Multi-RelSC` graphs from OssBuilds

## F.1 Metric Distributions

Figure 20 presents the degree distributions of the OssBuilds and Hadoop datasets. To enhance clarity and make patterns in the distributions more visible, the y-axis is displayed on a logarithmic scale. This adjustment highlights the spread of node degrees across a wide range, helping to capture variations that may be less noticeable on a linear scale.

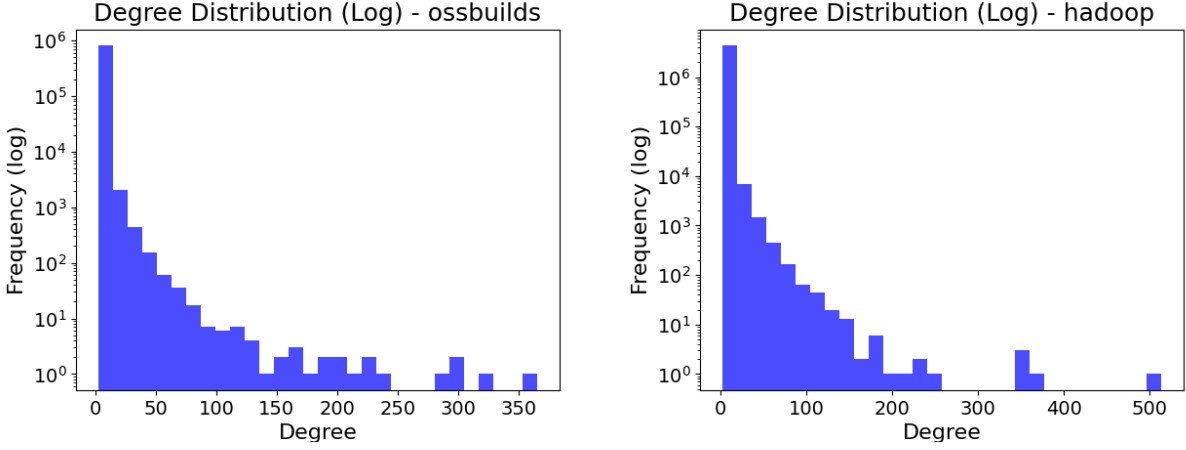

Figure 20: Degree distributions of OssBuilds (left) and Hadoop (right)

# G   Dataset Diversity and Bias Mitigation

To address concerns about the quality and representativeness of our dataset, we provide a detailed analysis of the diversity of code samples and the steps taken to mitigate potential biases in the data collection process. Our dataset comprises code from five distinct open-source projects collected through two different sources and methods, ensuring a broad coverage of code patterns and complexities relevant to software performance prediction tasks.

## G.1   Diversity of Code Samples

Our dataset includes code from the following projects:

- **OSSBuilds Dataset:**  This dataset encompasses four open-source projects, each contributing unique code patterns due to their different functionalities:

  - **SystemDS:** An Apache machine learning system for the data science lifecycle.
  - **H2:** A Java SQL database engine.
  - **Dubbo:** An Apache remote procedure call (RPC) framework.
  - **RDF4J:** A framework for scalable RDF data processing.

  These projects introduce a variety of code patterns, including database management, machine learning algorithms, RPC mechanisms, and data processing workflows. The diversity is reflected in the structural variations of the code and the resulting graphs.

- **HadoopTests Dataset:** Derived from the Apache Hadoop framework, this dataset includes 2,895 test files. Hadoop is renowned for processing large datasets across distributed computing environments, contributing complex code structures and control flows to our dataset.

Table 1 illustrates that the average number of nodes in the HadoopTests dataset is almost double that of the OSSBuilds dataset (1,490 vs. 875 nodes), indicating higher complexity in the Hadoop code samples. This indicates that our dataset has two main characteristics: the diversity of the code patterns and the complexity.

## G.2   Mitigation of Potential Biases

To minimize biases in our data collection process, we employed two different methods and environments:

- **OSSBuilds Data Collection:** Execution times were collected from the continuous integration (CI) systems of the respective projects using GitHub's shared runners. This approach leverages a standardized environment provided by the CI infrastructure, reducing variability due to hardware differences.

- **HadoopTests Data Collection:** We conducted multiple executions of Hadoop's unit tests on dedicated virtual machines within our private cloud. Each VM was configured with two virtual CPUs and 8 GB of RAM, and all non-essential services were disabled to ensure consistent performance measurements.

By diversifying our data sources and controlling the execution environments, we mitigated potential biases related to hardware configurations, workload fluctuations, and environmental inconsistencies.

## G.3   Representativeness and Generalization

The inclusion of diverse projects with varying functionalities enhances the representativeness of our dataset. The code samples encompass different structures, control flow statements, and data dependencies, which are critical for modelling software performance. The resulting graphs are generalized to various coding

patterns, excluding interface files that primarily contain function declarations without executable code. We intentionally did not include call graphs in the augmentation of ASTs to focus on the executable aspects of the code, which are more indicative of performance characteristics.

# H  Target values distributions

In this section, we present the distribution of target values for SystemDS, H2, Dubbo, and RDF4J, which are subprojects of OssBuilds. The distributions are shown in Figure 21.

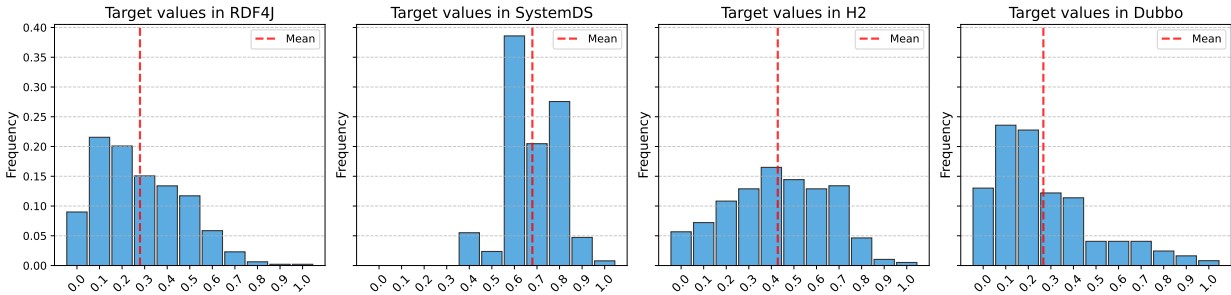

Figure 21: Distribution of target values for SystemDS, H2, Dubbo, and RDF4J, subprojects of OssBuilds.

