# OpenReview forum: "Broadening the Scope of Graph Regression: Introducing A Novel Dataset with Multiple Representation Settings"
_TMLR — Rejected by TMLR_

### Review · Reviewer_xumr · 2025-01-29

**Summary Of Contributions:**

The authors of this paper introduce RelSC and Multi-RelSC, two new datasets for graph regression in software performance prediction. In these datasets, the goal is to predict execution time from Java source code representations. The authors leverage Abstract Syntax Trees (ASTs), Control Flow Graphs (CFGs), and Data Flow Graphs (DFGs) to construct graph representations and benchmark several GNN models. The paper identifies a gap in graph regression datasets and addresses it by introducing a dataset from software engineering, a domain underexplored in graph machine learning.

**Audience:**

Yes

**Claims And Evidence:**

Yes

**Requested Changes:**

In its current form, I am leaning toward rejection, as the paper does not provide sufficient justification for its contribution.
The authors should include experiments where their model is used in a practical software engineering setting, they should include non-graph-based baselines, and they should properly introduce related work in this task.

**Strengths And Weaknesses:**

## Strong Points

- The paper introduces two new datasets for graph regression, which is a valuable resource for graph representation learning in the domain of software engineering.

- The authors clearly define how they construct the graphs from source code, and the methods are well documented. They also provide the source code.

- The authors test several GNN architectures, including GCN, GIN, GraphSAGE, and heterogeneous GNNs.


## Weak Points:

- My main concern lies in the limited motivation of the paper, as it fails to justify why execution time prediction from graph-structured code representations is a critical or impactful task. While the authors mention that their datasets could be useful for tasks like code optimization and performance-aware scheduling, they do not provide real-world use cases or concrete demonstrations of how their models could improve software engineering workflows. I believe that the paper would be strongly improved if the authors could incorporate such examples.

- Furthermore, the evaluation is limited to GNN-based models, with no comparison against traditional machine learning methods or non-graph deep learning approaches. Without such baselines, it remains unclear whether graph-based methods are necessary for execution time prediction or if simpler, more established techniques could perform just as well or better. The related work section is only referring to other graph datasets, without mentioning previous works on estimating the execution time of source code.

---

> ### Author Response · Authors · 2025-02-19
>
> We are pleased that the reviewer recognizes the importance of the proposed dataset, and we are glad that the reviewer found the method well-documented and noted that several GNN architectures were tested.
>
> Below we report a direct answer for each of your concerns.
>
> > W1: My main concern lies in the limited motivation of the paper, as it fails to justify why execution time prediction from graph-structured code representations is a critical or impactful task. While the authors mention that their datasets could be useful for tasks like code optimization and performance-aware scheduling, they do not provide real-world use cases or concrete demonstrations of how their models could improve software engineering workflows. I believe that the paper would be strongly improved if the authors could incorporate such examples.
>
> A1: We thank the reviewer for their valuable feedback. To better motivate the impact of execution time prediction, we have expanded Section 7 (Real-World Applications) by incorporating concrete examples:
> - Code Optimization: We now reference Facebook’s TAO system and Google Chrome’s optimization strategies, which rely on execution time predictions for performance tuning.
> - CI/CD: We highlight how large-scale CI/CD platforms (e.g., Microsoft, Netflix) use performance regression testing and how execution time estimation aids in detecting inefficient code changes.
> - Performance-Aware Scheduling: We clarify its role in cloud computing, where AWS Lambda and Google Cloud Functions depend on execution time predictions for resource allocation.
> These additions strengthen our paper’s motivation by demonstrating real-world applications of execution time estimation in software engineering.
>
>
> > W2: Furthermore, the evaluation is limited to GNN-based models, with no comparison against traditional machine learning methods or non-graph deep learning approaches. Without such baselines, it remains unclear whether graph-based methods are necessary for execution time prediction or if simpler, more established techniques could perform just as well or better. The related work section is only referring to other graph datasets, without mentioning previous works on estimating the execution time of source code.
>
> A2: We sincerely appreciate the reviewer’s insightful feedback. In response, we have made the following improvements to our work:
> - Expanded Related Work Section: We have revised the related work section to include two separate paragraphs—one discussing GNN-based approaches in software engineering and another covering traditional machine learning and non-graph deep learning methods. This ensures a more comprehensive overview of prior work, including studies on execution time estimation for source code.
> - Comparison with Non-Graph-Based Models: To empirically assess the necessity of graph-based representations, we have expanded our evaluation to include two additional baselines: Code2Vec and CodeBERT. Code2Vec is a transformer-based model that processes the abstract syntax tree, while CodeBERT treats source code as sequential text. Both models were trained and evaluated under the same conditions as our GNN-based methods. Our results show that these non-graph-based models consistently underperform compared to graph-based approaches, reinforcing the importance of explicitly modeling the structural dependencies within code.
>
> These additions provide a more balanced perspective and clarify the advantages of GNNs for execution time prediction. Thank you again for your valuable suggestions!

---

### Review · Reviewer_zWsu · 2025-01-30

**Summary Of Contributions:**

The paper introduces a new graph regression benchmark dataset aimed at predicting the runtime of software.
It proposes a unified graph representation of source code that integrates both syntactic (tree-based) and semantic (graph-based) representations.
Two versions of this graph representation are proposed and studied: RelSC (with node features) and Multi-RelSC (with multi-relational edges).
In an experimental comparison, various Graph Neural Network models are evaluated on these datasets, showing that homogenous GNNs trained on RelSC currently outperform more complex heterogenous GNNs trained on Multi-RelSC.

**Audience:**

Yes

**Claims And Evidence:**

Yes

**Requested Changes:**

The addition of more information about the target values and data splits is critical.

The paper would further be improved by a clearer presentation of experimental results and a discussion of alternative ways to predict runtime for source code.

Finally, it would be very helpful to provide a discussion on whether or not the current predictive performance of GNNs is practically useful if these models were used in a real software testing workflow. If it is currently not practically useful, is there a target error that would have to be reached in the future for this to be the case? Are there other ways to measure progression towards practically useful GNNs in this domain?

**Strengths And Weaknesses:**

Strengths:
* The task of software performance prediction is novel in the context of graph regression benchmarks and has practical relevance.
* The effect of how the raw data is represented as a graph is studied in detail, which is an important question that is commonly neglected in the graph learning literature.


Weaknesses:
* *Missing information about target values*: The paper provides insufficient details about the distribution of target values. What is their range? Are they normalized in some way or simply the raw runtime? What unit of time is used?
* *Missing information about experimental setup and data splits*: How the data is split into train/val/test splits is not discussed in detail. Are there fixed splits provided? What is the size of each split? Does the OSSBuild dataset consist of one training dataset that merges all 4 projects, or are there 4 separate training datasets and models are trained for individual projects?
* *Missing discussion of alternatives for predicting runtime*: Given the domain is source code, predictions of runtime could also naturally be obtained by a sequence model applied directly to the source code. Given the current success of LLMs as coding tools, it is not clear that GNNs can outperform such models in terms of prediction quality. While benchmarking GNNs is the focus of this work, such alternatives should still be discussed to justify where GNNs would have a practical advantage in this domain.
* *Presentation*: I found the presentation of results for Multi-RelSC in Section 6.2 a bit confusing: The first paragraph states that *"The
multi-relational nature of the Multi-RelSC datasets allows HeteroGAT to leverage these complexities for
better prediction accuracy"*. However, its error is worse than that of the simple homogenous GNNs trained on RelSC, which seems contradictory. The simpler models outperforming more complex GNNs is a very interesting result, but it should be presented more clearly.

---

> ### Author Response · Authors · 2025-02-19
>
> We are pleased that the reviewer acknowledges the novelty and practical relevance of the proposed dataset, as well as the thorough reporting of its data construction.
>
> Below we report a direct answer for each of your concerns.
>
> > W1:  Missing information about target values: The paper provides insufficient details about the distribution of target values. What is their range? Are they normalized in some way or simply the raw runtime? What unit of time is used?
>
> A1: We thank the reviewer for their valuable feedback. To clarify the details regarding target values, we emphasize that they are normalized between 0 and 1 for consistency across datasets. To improve clarity, we have added a new subsection (Section 5.2) where we explicitly describe the target value distributions for the main datasets, OssBuilds and Hadoop, which are now reported in Figure 6. Additionally, we provide the same analysis for the remaining datasets (SystemDS, H2, Dubbo, and RDF4J) in Appendix H.
>
> > W2: Missing information about experimental setup and data splits: How the data is split into train/val/test splits is not discussed in detail. Are there fixed splits provided? What is the size of each split? Does the OSSBuild dataset consist of one training dataset that merges all 4 projects, or are there 4 separate training datasets and models are trained for individual projects?
>
> A2: Thank you for your comment. We provide predefined train (70%), validation (15%), and test (15%) splits for all datasets. Each OssBuilds subproject (SystemDS, H2, Dubbo, and RDF4J) follows the same partitioning, with its splits fully contained within the corresponding sets of the full OssBuilds dataset. This ensures consistent evaluation at both the project-specific and dataset-wide levels, enabling fine-grained analysis and comparability. To clarify this in the main text, we add a paragraph in Section 8.
>
> > W3: Missing discussion of alternatives for predicting runtime: Given the domain is source code, predictions of runtime could also naturally be obtained by a sequence model applied directly to the source code. Given the current success of LLMs as coding tools, it is not clear that GNNs can outperform such models in terms of prediction quality. While benchmarking GNNs is the focus of this work, such alternatives should still be discussed to justify where GNNs would have a practical advantage in this domain.
>
> A3: We thank the reviewer for the comment. In response, we have expanded the discussion of related work to include how LLMs are used in software-related tasks. Additionally, we compare the GNN-based method with two other models: Code2Vec and CodeBert. Code2Vec is a transformer-based model that processes the abstract syntax tree, while CodeBERT treats source code as sequential text. Both models were trained and evaluated under the same conditions as our GNN-based methods. Our results show that these non-graph-based models consistently underperform compared to graph-based approaches, reinforcing the importance of explicitly modeling the structural dependencies within code.
>
> > W4 Presentation: I found the presentation of results for Multi-RelSC in Section 6.2 a bit confusing: The first paragraph states that "The multi-relational nature of the Multi-RelSC datasets allows HeteroGAT to leverage these complexities for better prediction accuracy". However, its error is worse than that of the simple homogenous GNNs trained on RelSC, which seems contradictory. The simpler models outperforming more complex GNNs is a very interesting result, but it should be presented more clearly.
>
> A4: Thank you for your feedback. While multi-relational graphs provide richer information, leveraging this complexity effectively remains a challenge. HeteroGAT benefits from the additional structure but does not outperform simpler models, making the proposed dataset a valuable resource for further research.
> We have clarified this point in Section 6.2 and the novel Section 6.3 (called discussion), emphasizing that simpler models outperform more complex ones. We appreciate your suggestion, which has helped improve the presentation of our findings.

---

> ### Author Response · Authors · 2025-02-19
>
> > Requested Changes 1: The addition of more information about the target values and data splits is critical.
>
> We thank the reviewer for their valuable feedback. The information regarding target values are now reported in a new subsection (Section 5.2), while data  split informations are reported in a new paragraph in Section 8.
> The paper would further be improved by a clearer presentation of experimental results and a discussion of alternative ways to predict runtime for source code.
>
>
> > Requested Changes 2: Finally, it would be very helpful to provide a discussion on whether or not the current predictive performance of GNNs is practically useful if these models were used in a real software testing workflow. If it is currently not practically useful, is there a target error that would have to be reached in the future for this to be the case? Are there other ways to measure progression towards practically useful GNNs in this domain?
>
> We appreciate the reviewer’s insightful question. Defining a precise target error for practical usability is challenging, as it depends on various factors specific to real-world software testing workflows. However, to better contextualize our results, we have added a new Discussion section (Section 6.3). This section examines the correlation achieved by the best-performing model (PNA) and highlights the presence of significant outliers, which could impact its reliability in real-world applications.
> While the current predictive performance may not yet be sufficient for direct deployment, our dataset provides a valuable benchmark for future research aimed at improving GNN models in this domain.

---

### Review · Reviewer_X7T4 · 2025-02-06

**Summary Of Contributions:**

In the submitted manuscript, the authors propose a framework which enables the representation of source code as attributed graphs. They furthermore, construct two graph regression datasets in which Java code is represented as graphs and the associated learning task is to regress execution time of the represented codes. Finally, the authors benchmark several GNNs on their proposed datasets.

**Audience:**

No

**Broader Impact Concerns:**

I do not have any concerns about this work that would necessitate a broader impact statement.

**Claims And Evidence:**

No

**Requested Changes:**

1] From your paper it is not quite clear how your work could be distinguished from the contributions made in earlier published work. In particular, you state in Section 1 that your contributions are 1) a unified framework for the conversion of source code to graph data and 2) the curation of two novel datasets. However, your unified framework seems to heavily rely on the definition of ASTs that were published in SAmoaa et al (2023a). And also with regards to contribution 2) you state in Section 4 that the dataset you publish here was "initially utilized in Samoaa et al. (2022a)" and you refer to them for the detailed description of the dataset construction ("For detailed instructions on constructing a RelSC graph, please refer to Samoaa et al. (2022a)."). So, it is unclear to me how your dataset is a novel contribution if it was already used to benchmark performance in published work three years ago. Further explanation of how your contributions extend previous work would be helpful to the reader.

2] In general the writing and clarity of the ideas discussed in this paper should be improved.

2.1] For example in Sections 3.2 and 3.3 you present control flow graphs and data flow graphs. These graph structures are provided without any reference and introduced in 6 and 5 lines, respectively. Since your proposed framework for representing source code as graphs builds on these two graphs, it seems necessary that you more carefully introduce these graph constructions at a higher level of abstraction than you currently do. Currently, you heavily rely on examples to introduce these frameworks, and while examples are nice to illustrate concepts, it is far clearer if these concepts are also defined explicitly.

2.2] Another example in which further clarification is necessary is in the caption of Figure 2, where you write "Package declarations, import statements, as well as the declaration in Line 15 are skipped for brevity." It is unclear whether this statement only applies to the example in Figure 2 or to your construction framework in general.

2.3] In parts the writing is somewhat repetitive. For example, the paragraphs starting with "RGCNs Schlichtkrull et al. (2017) extend GNNs to handle multi-relational graphs [...]" and "Multi-relational GNNs, such as Relational Graph Convolutional Networks (Schlichtkrull et al., 2017), [...]" on Page 4 seem to discuss near identical information.

3] Several of the claims in Section 4 are not sufficiently specific to be true in general. For example in the following claim "For our experiments, we employed two different real-world datasets of performance measurements to ensure robustness." it is unclear to me how you define the 'robustness' of a dataset and how you are able to ensure it by using two datasets.

4] You state that your dataset has 72 node types, which seems like quite a lot given the limited average size of your graphs. Furthermore in Figures 10-15, we observe that often edge relations are present less than 10 times on average in many datasets. Are you confident that different node and edge types are sufficiently frequently represented in your dataset for a model to learn any signal specific to the different types?

5] In the paragraph titled "Multi-RelSC" in Section 5 you discuss the "structural complexity" and "diversity" of this dataset. However, in the text you mainly discuss average node and edge counts. I am not sure whether these counts and their standard deviation are good indicators of structural complexity and diversity. I recommend that the authors either adjust their claims or explore how graph kernels or other methodology may allow them to quantify, measure and compare structural complexity and diversity of their datasets. The authors furthermore claim that this data set allows for "testing GNNs across different scenarios and domains", which seems erroneous since this dataset seems to originate in a single domain, which is the representation of source code as graphs.

6] In Section 6.2 you observe that "the choice of GNN architecture has a modest impact on performance for RelSC datasets." To me, it seems that a dataset on which the performance of all studied GNNs is identical may not be well-suited for the comparison and further development of GNNs. I therefore want to suggest that you extend your study to more recent GNN architectures in the hope that significant performance differences will arise which indicate that certain choices in the GNN models enable better performance on your proposed dataset.

7] Minor Comments:

7.1] Typos: throughout the paper you refer to "Figure 1" as "Listing 1"; "marking the graph more connected"; "SDT" instead of "STD"

7.2] In Equation (3) you use the set of neighbours as the upper limit of the summation operation. This is uncommon. I am more used to seeing integer valued quantities used as these limits.

7.3] In Section 4.3 you present 10 edge types that you add (the alphabetic list starts at (b) and in the text you claim to have 11 edge types. So, I worry type (a) may have been forgotten). However, these edge types are rather briely introduced. Potentially providing some examples of graphs in which these edge types are added would help the reader understand these edge types better.

7.4] In Section 4.4 you write "concatenated with the sum of its outgoing edge types". It is a bit unclear to me how these types can be summed.

7.5] In Section 6.1 I noticed that the years of publication of the GIN and GAT model appear to be slightly wrong (and to be different than in your list of references).

7.6] In Section 6.1 you state that the hidden dimension of your GNNs was fixed at 30 and the learning rate was set to 0.01. I was expecting a larger hidden dimension and a lower learning rate, which leads me to wonder whether you optimised these hyperparameters and if so which grid of possible hyperparameter values you considered in your study?

7.7] In Table 3 it is unclear why you studied the GAT model only for one of your two datasets. It would furthermore be easier to get an overview of your results if you highlighted the best performance in each setting by typesetting the accuracy in bold.

**Strengths And Weaknesses:**

Strengths:

1] The general point that you make on possible drawbacks of overstudying benchmark datasets from a small number of domains, mainly chemistry, is valid and indeed warrants further study of datasets from other domains to ensure that methodological progress on Graph Neural Networks does not overfit to any one given domain inadvertently.

2] I liked the way in which you used an example in Figure 1 to illustrate your graph construction.

3] Your ablation study in Table 4 is interesting.

Weaknesses:

1] The difference between your framework to represent source code as graphs and earlier frameworks that you reference, i.e., Samoaa et al. (2022a), is insufficiently explained.

2] One of the two main contributions of this paper is the curation of two datasets. However, these datasets contain 922 and 2895 graphs, which is probably lower than the median size of graph level datasets considered in the Graph Representation Literature. It is unclear to what degree datasets of such small size enable conclusive statements to be made about the studied domains. It furthermore seems that all studied GNNs exhibit near-equal performance on one of your datasets, which makes it unclear to what degree this dataset is suitable for the comparison of GNNs.

3] It seems to me that several aspects of your work need to be further specified and resolved for publication (see my Requested Changes).

---

> ### Author Response · Authors · 2025-02-19
>
> We appreciate the reviewer’s positive feedback on the main points of our paper, the example in Figure 1 and the ablation study.
>
> Below we report a direct answer for each of your concerns.
>
> > w1: The difference between your framework to represent source code as graphs and earlier frameworks that you reference, i.e., Samoaa et al. (2022a), is insufficiently explained.
>
> We sincerely thank the reviewer for the comment, we provide a detailed response bellow (Question 1 in Requested Changes)
>
> > w2: One of the two main contributions of this paper is the curation of two datasets. However, these datasets contain 922 and 2895 graphs, which is probably lower than the median size of graph level datasets considered in the Graph Representation Literature. It is unclear to what degree datasets of such small size enable conclusive statements to be made about the studied domains. It furthermore seems that all studied GNNs exhibit near-equal performance on one of your datasets, which makes it unclear to what degree this dataset is suitable for the comparison of GNNs.
>
> We sincerely thank the reviewer for their valuable feedback. We would like to clarify that while many publicly available graph-level datasets in the literature are indeed large, several widely studied real-world datasets contain fewer than 2,000 graphs. Examples include $O_2$Perm (595 graphs), ESOL (1,128 graphs), MUTAG (188 graphs), PTC (344 graphs), PROTEINS (1,113 graphs), ENZYMES (600 graphs), D&D (1,178 graphs), IMDB-BINARY (1,000 graphs), and IMDB-MULTI (1,500 graphs), among others. These datasets have been extensively used in evaluating Graph Neural Networks, demonstrating that smaller-scale datasets still provide meaningful insights into graph representation learning.
> Despite their relatively small size, datasets of this scale provide meaningful insights, particularly in evaluating GNN performance in low-data regimes, a scenario that is common in real-world applications where labeled data is often limited. Understanding how models perform in low-data settings is crucial for developing GNNs that generalize effectively beyond large-scale benchmarks.
> We appreciate the reviewer’s insightful comments, which have allowed us to further strengthen the discussion in our manuscript.
> The concern about the fact  that all studied GNNs exhibit near-equal performance is discussed later (answer 6 in Requested Changes)

---

> > ### Author Response · Authors · 2025-02-19
> >
> > > Requested Changes1 : From your paper it is not quite clear how your work could be distinguished from the contributions made in earlier published work. In particular, you state in Section 1 that your contributions are 1) a unified framework for the conversion of source code to graph data and 2) the curation of two novel datasets. However, your unified framework seems to heavily rely on the definition of ASTs that were published in SAmoaa et al (2023a). And also with regards to contribution 2) you state in Section 4 that the dataset you publish here was "initially utilized in Samoaa et al. (2022a)" and you refer to them for the detailed description of the dataset construction ("For detailed instructions on constructing a RelSC graph, please refer to Samoaa et al. (2022a)."). So, it is unclear to me how your dataset is a novel contribution if it was already used to benchmark performance in published work three years ago. Further explanation of how your contributions extend previous work would be helpful to the reader.
> >
> >
> > We appreciate the reviewer’s thoughtful feedback and the opportunity to clarify the novelty of our contributions. Below, we address the key concerns regarding the distinction of our work from prior research, specifically in relation to Samoaa et al. (2022a) and Samoaa et al. (2023a).
> >
> > ## Distinction from Samoaa et al. (2022a)
> > While Samoaa et al. (2022a) introduced a dataset, its scope was significantly more limited compared to the dataset presented in our work. The key distinctions are as follows:
> >  - **Enhanced Feature Representation:** The dataset in Samoaa et al. (2022a) did not include node features, whereas our dataset explicitly integrates node features that capture the semantic information extracted from the AST.
> >  - **Multi-Relational Graphs:** The dataset in Samoaa et al. (2022a) treated source code as a single-relation graph, limiting the diversity of relationships captured. In contrast, we introduce a multi-relational version (Multi-RelSC), where nodes are connected by multiple edge types representing different relationships. Multi-relational graph datasets for graph regression are rare in the literature, making our contribution particularly novel and valuable for future research.
> > - **Extended Scope and Utility:** Our dataset is not merely a reuse of the prior dataset but a significant extension and enhancement that provides more detailed and diverse representations of source code. This makes it a substantially more general and powerful resource for benchmarking GNN-based models in software engineering applications.
> > To ensure this distinction is clear to the reader, we explicitly clarify these enhancements in the manuscript.
> >
> > ## Clarification Regarding Samoaa et al. (2023a)
> > We would also like to emphasize that our work does not claim ASTs as a novel contribution. ASTs are a well-established representation for source code, initially introduced by McCarthy (1960) in the context of symbolic computation. Our work builds on this foundation, but our contribution lies in how we process and extend ASTs into relational and multi-relational graphs.
> > - Difference in Research Focus: Samoaa et al. (2023a) specifically investigates Active Learning techniques on ASTs (thus on trees), whereas our work focuses on graph regression tasks and the development of datasets that enhance GNN-based learning for source code analysis. These are fundamentally different research directions.
> > - Empirical Justification: As demonstrated in our ablation study (Section 6.4), applying GNNs directly to ASTs leads to inferior performance compared to using our proposed RelSC dataset. This empirical evidence underscores the advantage of structuring source code as relational and multi-relational graphs rather than treating it as a tree structure.
> >
> > To avoid any ambiguity, we refine our manuscript to explicitly differentiate our contributions from Samoaa et al. (2023a) and reinforce the unique aspects of our approach.
> >
> > McCarthy, John. "Recursive functions of symbolic expressions and their computation by machine, part I." Communications of the ACM 3.4 (1960): 184-195.

---

> > > ### Author Response · Authors · 2025-02-19
> > >
> > > > Requested Changes  2.1 and 2.2: In general the writing and clarity of the ideas discussed in this paper should be improved. For example in Sections 3.2 and 3.3 you present control flow graphs and data flow graphs. These graph structures are provided without any reference and introduced in 6 and 5 lines, respectively. Since your proposed framework for representing source code as graphs builds on these two graphs, it seems necessary that you more carefully introduce these graph constructions at a higher level of abstraction than you currently do. Currently, you heavily rely on examples to introduce these frameworks, and while examples are nice to illustrate concepts, it is far clearer if these concepts are also defined explicitly.
> > > Another example in which further clarification is necessary is in the caption of Figure 2, where you write "Package declarations, import statements, as well as the declaration in Line 15 are skipped for brevity." It is unclear whether this statement only applies to the example in Figure 2 or to your construction framework in general.
> > >
> > > We thank the reviewers for their valuable feedback. In response, we have revised Sections 3.2 and 3.3 to provide a clearer and more comprehensive explanation of Control Flow Graphs (CFGs) and Data Flow Graphs (DFGs). The figures have been better integrated with their corresponding explanations, and we have added a detailed breakdown of how CFGs and DFGs are constructed from the code example in Listing 1. Additionally, we have introduced both concepts at a higher level of abstraction to improve clarity. Finally, we have refined the figure captions, removing unnecessary details that could complicate understanding.
> > >
> > > > Requested Changes  2.3: In parts the writing is somewhat repetitive. For example, the paragraphs starting with "RGCNs Schlichtkrull et al. (2017) extend GNNs to handle multi-relational graphs [...]" and "Multi-relational GNNs, such as Relational Graph Convolutional Networks (Schlichtkrull et al., 2017), [...]" on Page 4 seem to discuss near identical information.
> > >
> > > We have carefully revised the manuscript to remove redundant sections and improve overall clarity. Thank you for pointing this out.
> > >
> > > >  Requested Changes 3. Several of the claims in Section 4 are not sufficiently specific to be true in general. For example in the following claim "For our experiments, we employed two different real-world datasets of performance measurements to ensure robustness." it is unclear to me how you define the 'robustness' of a dataset and how you are able to ensure it by using two datasets.
> > >
> > > We appreciate the reviewer’s insightful comment regarding the specificity of our claims in Section 4. To address this, we have revised the manuscript to clearly define the role of each dataset in our evaluation and removed the vague reference to "ensuring robustness." Instead, we now explicitly describe how the two datasets contribute to a broader evaluation—one dataset captures real-world variability in open-source projects, while the other provides a controlled experimental setting. This distinction helps clarify the motivation behind our dataset choices.

---

> > > > ### Author Response · Authors · 2025-02-19
> > > >
> > > > > Requested Changes 4. You state that your dataset has 72 node types, which seems like quite a lot given the limited average size of your graphs. Furthermore in Figures 10-15, we observe that often edge relations are present less than 10 times on average in many datasets. Are you confident that different node and edge types are sufficiently frequently represented in your dataset for a model to learn any signal specific to the different types?
> > > >
> > > > We sincerely thank the reviewer for their insightful comment. The statement, “..consisting of 72 unique node types,” in the AST Construction section refers to the node types generated by the Java parser (javalang). Specifically, code snippets are parsed into Abstract Syntax Trees (ASTs), where each node is assigned one of the 72 predefined node types (as illustrated in Figure 2). These node types are then used to construct node embeddings.
> > > > The 72 node types are not manually defined but are instead directly extracted from javalang, a widely used Java parsing library. These node types follow a standard representation in software engineering research. This ensures consistency and compatibility with existing methodologies in the field.
> > > >
> > > > Figures 10-15 depict relationships among categories in the Multi-relSC datasets, not between node types in the AST. AST node types are mapped to these categories based on their semantics, with Table 9 detailing the conversion from *NodeType* to *Category*. Specifically, node types are grouped into seven categories: *Declarations*, *Data Types*, *Control Flow*, *Operations*, *Structural Elements*, *Exceptions* and *Errors, and Others*.
> > > >
> > > > Finally, we would like to clarify that Figures 10-15 (renumbered as 12-17 in the revised manuscript) represent the average count of relations between categories. The low occurrence of certain relations is primarily due to two factors: (i) the relatively small graph sizes and (ii) the absence of specific relations in some graphs. However, rather than being a limitation, we believe this sparsity presents an exciting research opportunity. It highlights the challenge of learning meaningful representations from limited but structured relational data, which could inspire novel approaches in graph-based learning and model generalization.
> > > > In conclusion, we thank the reviewer for their comment. We have revised the text to make the distinction between node types and categories clearer.
> > > >
> > > >
> > > > > Requested Changes 5. In the paragraph titled "Multi-RelSC" in Section 5 you discuss the "structural complexity" and "diversity" of this dataset. However, in the text you mainly discuss average node and edge counts. I am not sure whether these counts and their standard deviation are good indicators of structural complexity and diversity. I recommend that the authors either adjust their claims or explore how graph kernels or other methodology may allow them to quantify, measure and compare structural complexity and diversity of their datasets. The authors furthermore claim that this data set allows for "testing GNNs across different scenarios and domains", which seems erroneous since this dataset seems to originate in a single domain, which is the representation of source code as graphs.
> > > >
> > > > We appreciate the reviewer’s valuable feedback regarding the characterization of structural complexity and diversity in the dataset. In response, we have adjusted our claims to more accurately reflect the information provided in the text.
> > > >
> > > > > Requested Changes 6. In Section 6.2 you observe that "the choice of GNN architecture has a modest impact on performance for RelSC datasets." To me, it seems that a dataset on which the performance of all studied GNNs is identical may not be well-suited for the comparison and further development of GNNs. I therefore want to suggest that you extend your study to more recent GNN architectures in the hope that significant performance differences will arise which indicate that certain choices in the GNN models enable better performance on your proposed dataset.
> > > >
> > > >
> > > > We appreciate the reviewer’s insightful comment. To further investigate the impact of GNN architectures on performance, we have extended our study by evaluating Principal Neighborhood Aggregation (PNA), a more recent and expressive GNN model.
> > > > Our results show that while the baseline models exhibit performance saturation, PNA achieves notable improvements, suggesting that architectural advancements can indeed differentiate models in this setting. This finding addresses the concern that all studied GNNs perform identically, demonstrating that the dataset is sensitive to model improvements and remains a valuable benchmark for assessing novel GNN architectures.

---

> > > > > ### Author Response · Authors · 2025-02-19
> > > > >
> > > > > Minor Comments:
> > > > >
> > > > > > Typos: throughout the paper you refer to "Figure 1" as "Listing 1"; "marking the graph more connected"; "SDT" instead of "STD"
> > > > >
> > > > > Thanks for spotting those typos! We've fixed them.
> > > > >
> > > > > > In Equation (3) you use the set of neighbours as the upper limit of the summation operation. This is uncommon. I am more used to seeing integer valued quantities used as these limits.
> > > > >
> > > > > Thanks, we’ve fixed equation (3).
> > > > >
> > > > > > In Section 4.3 you present 10 edge types that you add (the alphabetic list starts at (b) and in the text you claim to have 11 edge types. So, I worry type (a) may have been forgotten). However, these edge types are rather briely introduced. Potentially providing some examples of graphs in which these edge types are added would help the reader understand these edge types better.
> > > > >
> > > > > Edge type (a) corresponds to the original AST edges. We appreciate the reviewer for pointing this out and have now explicitly included type (a) in the list. Additionally, we have revised Section 4.3 to provide a more detailed description of all edge types.
> > > > >
> > > > > > In Section 4.4 you write "concatenated with the sum of its outgoing edge types". It is a bit unclear to me how these types can be summed.
> > > > >
> > > > > Thanks for your question! We concatenate the node features with the summed one-hot encodings of its outgoing edge types. This has been clarified in the manuscript
> > > > >
> > > > > > In Section 6.1 I noticed that the years of publication of the GIN and GAT model appear to be slightly wrong (and to be different than in your list of references).
> > > > >
> > > > > Thanks, we’ve fixed them.
> > > > >
> > > > > > In Section 6.1 you state that the hidden dimension of your GNNs was fixed at 30 and the learning rate was set to 0.01. I was expecting a larger hidden dimension and a lower learning rate, which leads me to wonder whether you optimised these hyperparameters and if so which grid of possible hyperparameter values you considered in your study?
> > > > >
> > > > > Thank you for your question! Our primary objective in this work is to introduce a novel dataset rather than optimizing model performance. To maintain consistency and focus on the dataset's utility, we used fixed hyperparameters without an extensive search. Future work could explore a more thorough hyperparameter optimization to assess its impact on model performance. We better clarify it in the main article.
> > > > >
> > > > > > In Table 3 it is unclear why you studied the GAT model only for one of your two datasets. It would furthermore be easier to get an overview of your results if you highlighted the best performance in each setting by typesetting the accuracy in bold.
> > > > >
> > > > > We report GAT results only for Multi-relSC to demonstrate the flexibility of our dataset in accommodating different architectures. Our primary focus was to showcase a diverse set of models rather than exhaustively evaluating each architecture on both datasets. In total, we tested seven different architectures on Multi-relSC, ensuring a comprehensive comparison. We appreciate the reviewer’s suggestion and have now highlighted the best-performing model in bold for improved clarity

---

### Author Response · Authors · 2025-02-19
**Summary of the main changes**

We thank all reviewers for their valuable comments and have directly addressed each question in our responses. We have uploaded a revised version of the manuscript, with all modifications in blue.

Here is a concise summary of the main changes:
 - We highlighted the importance of graph-based representations by introducing two additional models: Code2Vec, which is based on ASTs, and CodeBERT, which is based on raw source code. Our results show that these non-graph-based approaches do not achieve the same performance as our graph-based model, reinforcing the value of leveraging graph structures for execution time prediction.
- We integrated a more recent and advanced GNN-based architecture, PNA, demonstrating how state-of-the-art models further enhance performance.
- We expanded the discussion on real-world applications of execution time prediction, emphasizing its significance in software development.
- We included crucial details such as the validation split of the dataset and the distribution of target values.
- We incorporated a discussion on potential limitations of our current approach, particularly regarding the presence of outliers.
- We explicitly clarified the differences between our dataset and Samoaa et al.'s work.
- We improved the overall writing for better clarity and readability.

We strongly appreciate the received feedback, that allows us to improve our manuscript.

---

### Decision · Action_Editor_KehB · 2025-03-17

**Recommendation:** Reject

**Comment:**

Two reviewers still raise significant concerns about the present work after the revisions by the author. That is,
- The reviewers argue that the reported empirical results are not well "suited for the development and delineation of different architectural choices." For example, the reported empirical scores are very close for all architectures. The addition did not convince the reviewers of the new PNA architecture.
- The datasets are relatively small, and the reviewers were not convinced by the authors' response ("small-data regime") as many existing small benchmark datasets for graph learning exist.
The reviewers criticized the provided baselines for not being sufficiently tuned, which does allow for a good base for follow-up work.
- One reviewer remarked that too many changes were made to the manuscript during the revision, requiring a thorough review.

Overall, there was too much-remaining criticism to accept the paper, and the many changes required a complete review cycle to ensure their quality.

**Audience:**

Some individuals in TMLR's audience would be interested in the paper's findings.

**Claims And Evidence:**

Yes, the claims are mainly clear. However, the authors made significant changes during the rebuttal, justifying another full review cycle to ensure the quality of the changes. Moreover, some reviewers stress that the  empirical results are likely not "suited for the development and delineation of different architectural choices." For example, the reported empirical scores are very close for all architectures. The addition did not convince the reviewers of the new PNA architecture.

**Resubmission Of Major Revision:**

The authors may consider submitting a major revision at a later time.